# DiMSUM 🥟: Diffusion Mamba - A Scalable and Unified Spatial-Frequency Method for Image Generation

**Hao Phung**[*13†]     **Quan Dao**[*12†]     **Trung Dao**[1]
**Hoang Phan**[4]     **Dimitris N. Metaxas**[2]     **Anh Tran**[1]
[1]VinAI Research     [2]Rutgers University     [3]Cornell University     [4]New York University
htp26@cornell.edu     quan.dao@rutgers.edu     v.trungdt21@vinai.io
hvp2011@nyu.edu     dnm@cs.rutgers.edu     v.anhtt152@vinai.io

## Abstract

We introduce a novel state-space architecture for diffusion models, effectively harnessing spatial and frequency information to enhance the inductive bias towards local features in input images for image generation tasks. While state-space networks, including Mamba, a revolutionary advancement in recurrent neural networks, typically scan input sequences from left to right, they face difficulties in designing effective scanning strategies, especially in the processing of image data. Our method demonstrates that integrating wavelet transformation into Mamba enhances the local structure awareness of visual inputs and better captures long-range relations of frequencies by disentangling them into wavelet subbands, representing both low- and high-frequency components. These wavelet-based outputs are then processed and seamlessly fused with the original Mamba outputs through a cross-attention fusion layer, combining both spatial and frequency information to optimize the order awareness of state-space models which is essential for the details and overall quality of image generation. Besides, we introduce a globally-shared transformer to supercharge the performance of Mamba, harnessing its exceptional power to capture global relationships. Through extensive experiments on standard benchmarks, our method demonstrates superior results compared to DiT and DIFFUSSM, achieving faster training convergence and delivering high-quality outputs. The codes and pretrained models are released at https://github.com/VinAIResearch/DiMSUM.git.

## 1   Introduction

Diffusion models [58, 22] are a trending generative model technique that has gained significant attention in machine learning and computer vision. The core idea behind diffusion models is to learn how to reverse the diffusion process by gradually transforming a simple initial distribution, like Gaussian noise, into a complex data distribution. The flexibility, robust performance, and high-quality outputs make them powerful tools for advancing the state-of-the-art in generative modeling, and large diffusion-based generators have revolutionized the field of image [51, 3], video [23, 21], and 3D synthesis [50, 62, 61]. While diffusion models initially rely on UNet architectures, recent methods have shifted gear to build upon transformer backbones. A line of works [48, 14, 11] have shown that transformer-based diffusion models are scalable and consistently offer higher generation quality

---

[*]Equal contributions.

[†]Work done while at VinAI.

38th Conference on Neural Information Processing Systems (NeurIPS 2024).

than the UNet-based counterparts. Even the most common open-source text-to-image tool, Stable Diffusion, has switched to use transformers in their upcoming release [53]. Hence, transformers are becoming the new backbone standard for diffusion models. The power of this structure lies in the attention mechanism for capturing richer in-context relations. However, transformers have the drawback of a costly quadratic complexity, which might hinder their feasibility for high-dimensional data.

While transformers are taking over state-of-the-art diffusion backbones, a novel technique called state-space models (SSMs) has suddenly arrived, promising a better alternative. SSMs [18, 16, 15] have revolutionized the NLP field, favoring linear time complexity and excelling at long-context modeling. This type of network bears similarities to the recurrent process of RNNs while being capable of fully operating in parallel like convolutional networks. SSM promises to surpass transformers in most tasks, prioritizing compute efficiency, such as long-sequence modeling. Mamba [15] is a special type of SSM that offers greater quality by introducing time conditioning and context dependency to the hidden states. In the context of computer vision, within a very short period, this architecture has been used to address a variety of problems, including image perception [74, 28, 33], image restoration [20, 71], and image generation [41, 74, 65, 25]. In diffusion-based image generation, Diffusion State Space models (DIFFUSSM) [65] already surpass their transformer-based counterparts.

Though showing many advantages, Mamba still has a critical weakness when processing 2D imagery data. Like vision transformers, images are divided into patches and then mapped into tokens. Mamba processes these tokens following a specific scanning order, thus introducing an inductive bias about 2D images into the model. Specifically, this order greatly impacts the interplay between image tokens, thereby affecting model performance. This characteristic is unfavorable, particularly when transformers have no such order-dependency issue. Many vision-based Mamba studies have focused on solving this problem on proposed advanced scanning mechanisms like bi-directional [74], cross-scanning [41, 33], or 8-directions zigzag [25]. Despite improving performance, these scanning techniques still fail to capture global and long-range relations and do not fully release Mamba's potential.

In this paper, we enhance Mamba-based diffusion models, specifically focusing on image generation. Previous models failed to address the scanning order issue due to their exclusive reliance on spatial processing, overlooking crucial long-range relations that manifest in the frequency spectrum. We suggest a novel approach integrating frequency scanning with the conventional spatial scanning mechanism. Although initial work in Mamba has explored this combination for a limited task of image deraining [71], it lacked a comprehensive analysis of the effective integration of these features.

Motivated by the above observation, this paper introduces DiMSUM, a novel architecture that harnesses Mamba's power to unlock diffusion models' generation capabilities. Our approach enhances sensitivity to local structures and long-range dependencies by integrating wavelet transforms and spatial information. Using a query-swapped cross-attention technique, we dynamically synergize spatial and frequency information, accelerating convergence and improving image synthesis quality. Consequently, this boosts image quality and enhances the efficiency and scalability of the training.

Additionally, we incorporate globally shared transformer blocks to address global context integration, a limitation of traditional Mamba models. The block can also be viewed as a token-mixing layer that enriches global relations among image tokens, addressing the weak inductive bias of the manually defined scanning orders in the original Mamba. Hence, DiMSUM can maintain high performance even with larger, more complex datasets. Extensive experiments show that DiMSUM achieves state-of-the-art FID scores and recall, setting a new benchmark in generative image modeling.

In summary, our contributions lie three-fold: (1) A novel Mamba architecture for diffusion models that leverages both spatial and frequency features to enhance the awareness of local structures within input images, leading to better image generation. (2) We interleave globally shared transformer blocks per a certain number of Mamba blocks. The transformer with a strong capacity for capturing global relationships significantly boosts generation results when integrating with Mamba. The transformer can also be seen as an order-invariant mixing layer that complements Mamba's loose assumptions about the order of 2D data. (3) Superior results across image generation benchmarks like ImageNet, CelebA-HQ, and LSUN Church. Additionally, our method maintains comparable GFLOPs and parameters with existing diffusion architectures while offering faster training convergence.

## 2 Related Work

### 2.1 State Space Models and Their Applications in Vision Tasks

In control engineering and system identification, state space models (SSMs) are described by state variables and first-order differential equations but initially underperformed in deep learning. Recent enhancements, notably by S4 [18] through the use of a HiPPO [16] initialization matrix, have significantly improved SSMs. Subsequent studies [18, 15, 17, 35, 1] show that SSMs can match transformers in long-range sequence modeling with the added benefit of linear time and space complexities. Notably, Mamba [15] has advanced over transformers in NLP by using a time-varying system with context-dependent parameters, enhancing the differentiation of hidden states over long sequences. This positions Mamba as a strong alternative to transformers across various domains.

In computer vision, ViM and VMamba [74, 41] are the first works to introduce Mamba as a building block in discriminative tasks. Sequentially, Mamba is explored in many computer vision tasks such as medical imaging [43, 39], point clouds [70, 34], and image generation [25, 73]. Similar to vision transformers, images are divided into patches, and the patches are mapped into tokens. The tokens are then arranged in a sequence following a scanning order. In vision transformers, the scanning order does not matter since attention scores are computed between every token pair. However, SSMs consider the order information, introducing an inductive bias about 2D images into the model. Therefore, scanning order is vital in setting vision models' performance. ViM [74] proposed a bidirectional scanning order (sweep-2) for discriminative tasks. VMamba [41] proposed cross-scanning (sweep-4) per each Mamba building block. This cross-scan improves the model performance but costs enormous overhead. MambaND [33] reduces that cost by introducing two methods: interleaved scanning and multi-head scanning. Interleaved scanning, which alternates the scanning order in the sequential blocks, is simpler but gains better performance in discriminative tasks. Recently, Zigma [25] proposed a zigzag-8 scanning order to preserve the locality property (i.e., each token is adjacent to its next and previous tokens). The zigzag-8 scanning order shows faster convergence compared to the bidirectional one. In this paper, we show that too many scanning orders, e.g., sweep-8 and zigzag-8, may introduce excessive information and lead to worse performance than sweep-4. Instead, sweep-4 offers the best performance (Section 4.4).

### 2.2 Diffusion Architecture

Diffusion models [22, 57, 56, 51] are an emerging type of generative model that requires a sequential denoising process of several to thousands of steps to sample an image from initial Gaussian noise. Notably, most of them are based on Stochastic Differential Equations (SDE) that require an accumulation of additional stochastic noise at each generation step. Alternatively, there is a line of flow matching methods [36, 40, 44, 5] that emphasize deterministic trajectories from pure Gaussian noise to the target data distribution, favoring a straighter solution path. Their applications span across different tasks like image super-resolution [13], depth estimation [19], and motion synthesis [26]. Recent works [44, 31] have proved that diffusion models and flow matching are strongly correlated and can be converted into each other. In this work, we only focus on the simple objective of flow matching for our design.

Meanwhile, most methods are originally based on Unet architecture, which utilizes convolution resblock[†] to capture local information at multiscale resolution. The attention layer is also used, interleaving between resblock layers to capture global information. Recently, the vision transformer [9, 42] has emerged and largely surpassed CNNs in many tasks. For diffusion image modeling, several transformer-based architectures [48, 2, 14] have been recently introduced. Transformer-based architectures capture global information better than Unet ones and can generate higher-quality images. Specifically, UViT [2] replaced convolution resblock layers with transformer blocks and removed downsampling/upsampling blocks. DiT [48] directly replaced Unet with a vision transformer. Inspired by this, MDT [14] and MaskDiT [72] introduced a mask latent modeling approach to better capture contextual information and enhance training efficiency. Although transformer architectures achieve better image generation, these models suffer from quadratic time and memory complexity, slowing down training and inference processes. Recently, with the birth of Flash Attention [8, 7], both training and inference time of these transformer-based models are significantly reduced thanks to

---

[†] A residual block is a skip-connection block that learns residual functions relative to the layer inputs.

the reduced IO bottleneck. However, the computation time complexity remains quadratic. Recently, the S4 [18] model has been introduced to effectively deal with long-range dependency in the NLP field. Furthermore, the S4 model favors the linear complexity time and space, which is more efficient than the transformer. Among S4 class models, Mamba [15] stands out for its high performance in capturing long-range dependency. In diffusion models, DiffuSSM [65] adopts S4D [17] as a building block for their model and achieves better FID compared to transformer counterparts. Recently, Zigma [25] utilized Mamba for diffusion architecture, using a zigzag scanning order to preserve locality-aware scanning property. Despite showing promising results, Mamba-based diffusion models still struggle to find an optimal scanning scheme to take advantage of the 2D inductive bias from images. We find these approaches stuck in spatial processing, thus failing to incorporate global and local relations. These relations can effectively captured in frequency spectrum, thus we propose to incorporate frequency scanning alongside the existing spatial scanning mechanism.

## 2.3 Frequency-based networks

Employing frequency components extracted by Fourier, Cosine, or Wavelet transform in solving vision tasks was common in classical computer vision. Many modern works still find this practice useful in improving the performance of deep neural networks. In perception tasks, several works [37, 38, 67] integrate frequency processing in transformer architecture. NomMer [37] applied a discrete cosine transformer into global attention to efficiently yield synergistic context from both global and local contexts. To improve Masked Image Modeling, Ge-AE [38] introduces an additional frequency decoder using Fourier transform to reconstruct the high-frequency information better. Wave-Vit [67] applies wavelet into self-attention modules to reduce the time and space complexity of the transformer architecture while still preserving the performance. Recent work Simba [47] introduced Fourier-based layers (EinFFT) in combination with Mamba block to replace MLP layers for better channel mixing. To solve the image denoising problem, FreqMamba [71] applied a wavelet and Fourier transformer to process features injected into the Mamba block. In generative modeling, several works [66, 49, 69] corporate wavelet frequencies into generative framework. By explicitly decomposing features/images into high- and low-frequency bands through wavelet transform, the generative model can train stably and converge faster. Furthermore, the high frequencies can be learned more efficiently, leading to a sharper synthesis image. Observing the benefit of wavelet processing in generative modeling, we apply discrete wavelet transform on local features before feeding into the Mamba layers. By using cross-attention to fuse wavelet frequency features and spatial features, our method achieves significant improvement in image synthesis compared to merely spatial feature processing.

## 3 Method

This section presents DiMSUM, a novel architecture aiming for effective and high-quality image synthesis. Preliminary knowledge will be provided in section 3.1, then overview structure and mechanism of the proposed network (section 3.2), and finally its core components (sections 3.3 and 3.4).

## 3.1 Preliminary

**State Space Model (SSM).** SSM is a new type of sequence model that uses an implicit hidden state $h(t) \in \mathbb{R}^{N \times L}$ to map the $1D$ input signal $x(t) \in \mathbb{R}^L$ to its corresponding output signal $y(t) \in \mathbb{R}^L$. This process is formulated by a parameter matrix $\mathbf{A} \in \mathbb{R}^{N \times N}$ and two projection parameters $\mathbf{B} \in \mathbb{R}^{N \times 1}$ and $\mathbf{C} \in \mathbb{R}^{1 \times N}$:

$$h'(t) = \mathbf{A}h(t) + \mathbf{B}x(t), \quad y(t) = \mathbf{C}h'(t).$$

For practical usage, the continuous parameters $(\mathbf{A}, \mathbf{B})$ are discretized by a time-scale parameter $\mathbf{\Delta}$ to produce discrete parameters $(\overline{\mathbf{A}}, \overline{\mathbf{B}})$, following a zero-order hold (ZOH) rule:

$$\overline{\mathbf{A}} = \exp(\mathbf{\Delta} \cdot \mathbf{A}), \quad \overline{\mathbf{B}} = (\mathbf{\Delta} \cdot \mathbf{A})^{-1}(\exp(\mathbf{\Delta} \cdot \mathbf{A}) - \mathbf{I}) \cdot \mathbf{\Delta} \cdot \mathbf{B}.$$

Hence, the continuous system is rewritten as follows:

$$h_t = \overline{\mathbf{A}}h_{t-1} + \overline{\mathbf{B}}x_t, \quad y_t = \mathbf{C}h_t.$$

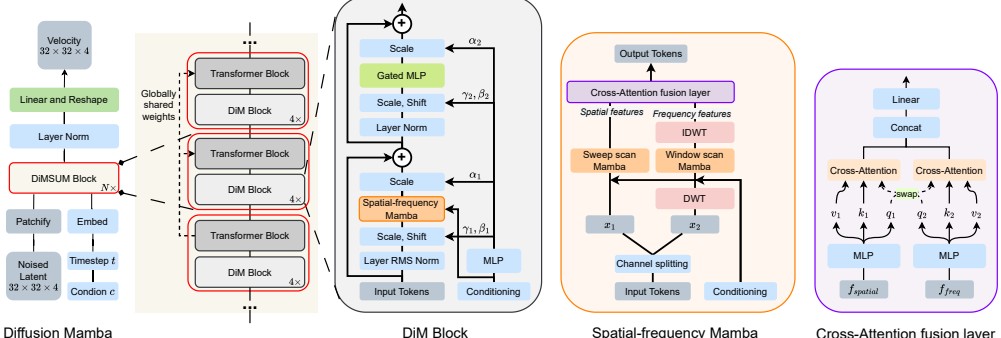

Figure 1: Overview of DiMSUM architecture.

Mamba then proposes a selective mechanism to enrich the dynamic interactions of different sequential states. In other words, the constant parameters $(\overline{\mathbf{B}}, \mathbf{C}, \mathbf{\Delta})$ are tuned into input-dependent parameters, enforcing the context awareness of input sequence states:

$$\overline{\mathbf{B}}_t = \text{Linear}_N(x_t), \quad \mathbf{C}_t = \text{Linear}_N(x_t), \quad \mathbf{\Delta_t} = \text{Softplus}(\text{Broadcast}_{\mathbf{L}}(\text{Linear}_{\mathbf{1}}(\mathbf{x_t}))),$$

where $\text{Linear}_*(.)$ is a projection layer to $*$-dimensional vector, $\text{Softplus}(.) = \log(1 + \exp(.))$, and $\text{Broadcast}_L(.)$ means duplicating a single-value vector to $L$-dimensional vector.

**Diffusion model.** Diffusion models [55, 22, 57, 58, 51] are also known as score-based models that learn the transitional trajectories from a Gaussian noise to signals in the target domain. Most methods are based on Stochastic Differential Equation (SDE), requiring a larger number of function evaluations to generate an image. Recently, Flow matching [36, 40, 5, 44] has proved to be a promising method that finds a deterministic mapping between input Gaussian noise and input data via solving Ordinary Differential Equation (ODE). Given an input data $x$ belonging to the modeling distribution $p(x)$ and a random noise $\epsilon \in \mathcal{N}(0, \mathbf{I})$, the forward process is formulated as:

$$x_t = x\alpha_t + \epsilon\sigma_t, \tag{1}$$

where $x_t$ is the noise-added data at a time step $t \in [0, 1]$ and $(\alpha_t, \sigma_t)$ are time-dependent functions of $t$. Particularly, these functions are constrained such that $\alpha_1 = \sigma_0 = 0$ and $\alpha_0 = \sigma_1 = 1$ to produce correct mapping between data $x$ at $t = 0$ and noise $\epsilon$ at $t = 1$. Specifically, [36, 40, 5] use a simple linear function where $\alpha_t = 1 - t$ and $\sigma_t = t$. We employ Flow matching's training objective to estimate the velocity between noise $\epsilon$ and data $x$:

$$\hat{\theta} = \underset{\theta}{\arg\min} \, \mathbb{E}_{t,x_t} \left[ ||x_1 - x_0 - v_\theta(x_t, c, t)|| \right], \tag{2}$$

where $v_\theta$ is a velocity estimator implemented by a neural network with parameters $\theta$ and $c$ is an input condition (e.g., class or text). If no condition is used, $c$ is set to empty.

**Wavelet transformation.** Among frequency transform techniques, wavelet transform stands out for its simplicity and efficiency. It preserves the structure of image space, with low-frequency subbands representing down-sampled approximations of the input image, while high-frequency ones emphasize local details such as vertical, horizontal, and diagonal edges. Particularly, Haar feature is the most prevalent wavelet transform, consisting of a low-pass filter $L = \frac{1}{\sqrt{2}} \begin{bmatrix} 1 & 1 \end{bmatrix}$ and a high-pass filter $H = \frac{1}{\sqrt{2}} \begin{bmatrix} -1 & 1 \end{bmatrix}$. To decompose an image $x \in \mathbb{R}^{H \times W}$, it needs to construct 4 kernels $LL^T, LH^T, HL^T, HH^T$, then applies them to the input image to extract corresponding subbands $\{x_{LL}, x_{LH}, x_{HL}, x_{HH} \mid x_* \in \mathbb{R}^{H/2 \times W/2}\}$, respectively. This process is called discrete wavelet transform (DWT). Notably, these filters are pairwise orthogonal, so an invertible matrix exists to map the data back to the original image space, coined as discrete inverse wavelet transform (IDWT). Given its benefits, we propose to use wavelet transform to supplement the local structure of frequency components into the process of Mamba, thus leading to enhanced image quality and training convergence, as demonstrated through our empirical experiments in section 4.

### 3.2 Overview of the proposed network

Inspired by the advancement of Mamba-based diffusion models and frequency-based networks, we design a novel architecture DiMSUM for effective and high-quality image synthesis with the structure

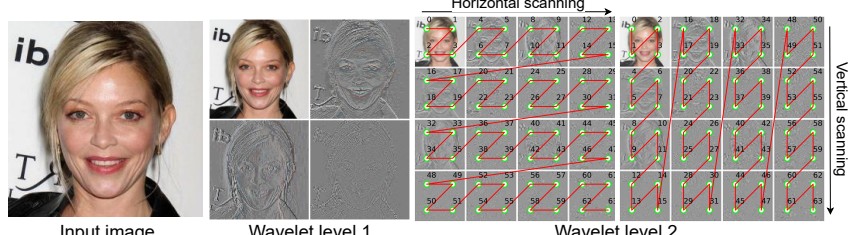

Figure 2: Illustration of Wavelet Mamba (Best view in color). For illustration purpose, we plot wavelet representations of an input image but our real process is performed on encoded features of the input. Giving an image of size $(8, 8)$, for example, it is first decomposed to four wavelet subbands of size $(4, 4)$ where each is further transformed to 2nd-level subbands of size $(2, 2)$. Green dots indicate pixel points within each wavelet subband and a window of size $2 \times 2$ is used to perform scanning across multiple wavelet subbands like the CNN kernel.

presented in Fig. 1. Similar to Latent Diffusion Models [51], our method performs image generation on the latent space of a pre-trained encoder [†]. The method first receives an input image and encodes it to a latent map of size $4 \times H \times W$. It then processes the latent map using our proposed Diffusion Mamba network, whose core is a sequence of DiMSUM blocks, each consisting of DiM blocks that employ a novel Mamba structure with spatial and frequency scanning fusion and a globally weight-shared transformer block. The processed latent is then decoded to the output image.

### 3.3 DiM block

A core component of our approach is the DiM block that relies on a novel Spatial-Frequency Mamba fusion technique. In this section, we will discuss in detail the ideas behind this vital component.

**Scanning in frequency space.** Mamba-based approaches in diffusion models often lack effective scanning schemes for preserving both local and global 2D spatial information. Although several works have proposed different heuristic scanning methods [74, 41, 33, 25] to address this issue, these approaches are insufficient for capturing local pixel dependencies and long-range frequency relationships. Though LocalMamba [28] proposed a window scanning to mimic the convolution kernel, it often underperforms compared to previously mentioned scanning methods as it is limited to the dependencies of nearby-pixels within window.

DiMSUM addresses these challenges by decomposing the original image into frequency wavelet subbands. This approach is effective to capture long-range frequency while preserving relations across different subbands. We redesigned the window scanning, where each window corresponds to a subband of the frequency space as in Fig. 3. Consequently, each window captures the full range of low/high-frequency signals from the original image. This advantage sets us apart from traditional window scanning in image space. As the model progresses through different subbands, it incorporates spatial information represented at various low-to-high frequencies, adding valuable context to the denoising process.

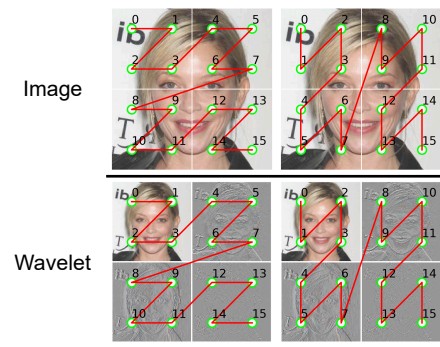

Figure 3: Comparison of window scanning on image and wavelet space. For illustration, one-level wavelet transformation is applied and each subband is half the resoluton of original image.

**Wavelet Mamba.** We now examine the integration of the wavelet transform into the Mamba structure shown in Fig. 2. Wavelet Mamba first applies DWT to decompose input features into wavelet subbands. Our main setting uses two-level Haar wavelet to map input into low and high-frequency features. Given input feature $x \in \mathbb{R}^{C \times H \times W}$, first-level wavelet transform is applied to produce 4 wavelet subbands of size $\mathbb{R}^{C \times H/2 \times W/2}$. Each

[†] https://huggingface.co/stabilityai/sd-vae-ft-ema

subband is then further decomposed into second-level wavelet subbands of size $\mathbb{R}^{C \times H/4 \times W/4}$. This feature is pivotal, as we decompose every wavelet subband to evenly separate an input image into multiple wavelet patches, unlike conventional wavelet transformations that process only LL subbands at the next level. In Wavelet Mamba, we concatenate those subbands to form a 1D sequence, apply window scanning within each subband, and slide across the sequence for feature extraction. The window scanning is inspired by a CNN kernel proposed in [28] with two window sliding directions: left to right and top to bottom. Note that since low-frequency subbands capture the main content of image, it should be input first. Therefore, we do not use reverse scanning orders: right to left and bottom to top. After passing through Wavelet Mamba, the output features are transformed back to input shape by using IDWT twice. With wavelet module, our model can better capture local structure of frequency information. Thus, incorporating Wavelet Mamba with Spatial Mamba can offer better performance, yielding high-quality image generation (Spatial-frequency Mamba in Fig. 1).

**Cross-Attention fusion layer.** Given $f_s$ and $f_w$ are spatial and wavelet features obtained by sweep and window scan. We combine these features using a cross-attention fusion layer as follows:

$$q_s, k_s, v_s = \text{Linear}(f_s), \qquad q_w, k_w, v_w = \text{Linear}(f_w),$$
$$f_{out} = \text{Linear}(\text{Concat}(\text{Attn}(q_s, k_w, v_w), \text{Attn}(q_w, k_s, v_s))).$$

More specifically, we first compute each feature's query ($q_*$), key ($k_*$), and value ($v_*$) using linear layers. To fuse the information between spatial and wavelet features, we do cross-attention by swapping the queries ($q_*$) of spatial and wavelet before applying a self-attention module onto each key, query, and value triplet. Finally, we concat the outputs of two cross attentions by channel followed by a linear projection to obtain the output feature $f_{out}$ (see the last subfigure in Fig. 1).

**Conditional Mamba**. Unlike attention modules, conventional Mamba has no explicit mechanism to inject input conditions into its flow. We propose a simple technique that enables Mamba to take in any conditional input $c$ via initializing the very first hidden state with the embedding $c$ instead of zero, as in original Mamba. This can be considered as a type of prior injection into Mamba. Specifically, the recurrent process of Mamba can be rewritten as below:

$$\begin{cases} h_0 & = \overline{\mathbf{A}} h_{-1} + \overline{\mathbf{B}} x_0 \\ y_0 & = \mathbf{C} h_0 \end{cases}, \qquad \begin{cases} h_t & = \overline{\mathbf{A}} h_{t-1} + \overline{\mathbf{B}} x_t \\ y_t & = \mathbf{C} h_t \end{cases} \tag{3}$$

In conventional Mamba, $h_{-1}$ is set to zero as there is no previous state at the beginning. Here, we set $h_{-1} = \text{Linear}_D(c)$ to inject context prior into flow of Mamba. As shown in ablation, conditional mamba effectively enhances model performance. This is beneficial for both unconditional and conditional generation. For unconditional image generation, we create an auxiliary learnable token to capture image space's global information, similar to vision transformers [10, 59]. For class-conditional generation task, we use a class embedding to condition on Mamba. Conditional Mamba is enabled by default in DiM blocks (Fig. 1).

### 3.4 Globally-shared attention block

Since Mamba is better than transformer at long-range dependency [18, 15] but weaker than transformer at in-context learning [46], we propose a hybrid mamba-transformer architecture which favors both these properties as in recent work Jamba [35]. Motivated by Zamba [1], we introduce the globally-shared transformer (attention) block. This shared attention block is added after each of four DiM blocks as shown in Fig. 1 since we want to preserve the continuity of the 4-sweep alternative scanning order. By using shared weights, we significantly reduce the number of parameters introduced by different attention blocks. This layer complements the flow of Mamba since transformers excel at extracting global relations without relying on manually defined orders of input sequences, as in Mamba. Hence, with this hybrid architecture, our method effectively addresses Mamba's order dependence while significantly reducing the FID with very few additional parameters.

## 4 Experiments

**Implementation.** We established a depth of 20 layers, a base width of 1024, and a patch size of 2 for network configuration (further information on hyperparameters in appendix A). We run experiments on standard datasets: CelebA-HQ[27], LSUN Church[68], and ImageNet[52]. For the sampling method, we follow [44, 5, 25] to use adaptive ODE solver 'dopri5' for evaluation. We assessed

| Model | NFE↓ | FID↓ | Recall↑ | Epochs |
|---|---|---|---|---|
| Ours | 61 | **4.62** | **0.52** | **225** |
| Zigma† [25] | 65 | 7.66 | 0.40 | 400 |
| LFM-8 [5] | 89 | 5.26 | 0.46 | 500 |
| LDM-4 [51] (ADM) | 500 | 5.11 | 0.49 | 600 |
| LDM-8 (ADM)‡ | 250 | 15.37 | - | 500 |
| LDM-8 (DiT)‡ | 250 | 10.21 | - | 500 |
| LSGM [60] | 23 | 7.22 | - | 1K |
| WaveDiff [49] | 2 | 5.94 | 0.37 | 500 |
| DDGAN [64] | 2 | 7.64 | 0.36 | 800 |
| RDUOT [6] | 2 | 5.60 | 0.38 | - |
| Score SDE [58] | 4000 | 7.23 | - | 6.2K |

(a) Size $256 \times 256$. † is our reproduced result and ‡ is adopted results from LFM paper.

| Model | NFE↓ | FID↓ | Recall↑ | Epochs |
|---|---|---|---|---|
| Ours | 82 | **6.09** | **0.46** | **165** |
| LFM-8 [5] | 94 | 6.35 | 0.41 | 500 |
| WaveDiff [49] | 2 | 6.40 | 0.35 | 400 |
| DDGAN [64] | 2 | 8.43 | 0.33 | 400 |

(c) Size $512 \times 512$.

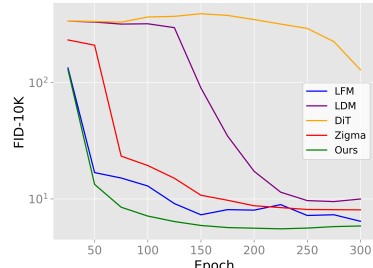

(d) Comparison of FID-10K over training epochs.

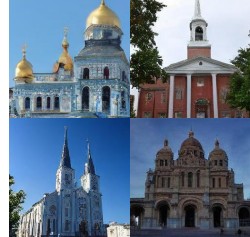

(b) Qualitative results.

Figure 4: Result of our model versus others upon CelebA-HQ. is our reproduced result based on Zigma's official code, and is an adopted result from LFM paper.

| Model | NFE↓ | FID↓ | Recall↑ | Epochs |
|---|---|---|---|---|
| Our | 73 | **3.76** | **0.56** | **395** |
| DIFFUSSM [65] | 250 | 3.94 | - | - |
| LFM-8 [5] | 90 | 5.54 | 0.48 | 500 |
| LDM [51] | 250 | 4.02 | 0.52 | 400 |
| WaveDiff [49] | 2 | 5.06 | 0.40 | 500 |
| DDPM [22] | 1000 | 7.89 | - | 640 |
| StyleGAN [30] | 1 | 4.21 | - | - |
| StyleGAN2 [29] | 1 | 3.86 | 0.36 | 13K |

(a) Quantitative results.

(b) Qualitative results.

Figure 5: Result of our model versus others upon LSUN Church $256 \times 256$ dataset.

performance using the Fréchet Inception Distance (FID)[45] and Recall[32] by first generating 50K images and then comparing them with the full reference dataset. We also report GFLOPs and the number of training iterations per second to further illustrate the model efficiency.

## 4.1 Image Generation

On the CelebA-HQ dataset at resolutions of 256x256 and 512x512 (Fig. 4), our method achieved state-of-the-art FID scores of 4.62 and 6.09, respectively, surpassing the scores reported in recent studies. Furthermore, DiMSUM demonstrated superior recall scores, indicating a greater diversity in the generated samples compared to other methods. This result is particularly impressive, given the majority of baseline methods are based on diffusion processes, which are known for excellent diversity. On LSUN Church (Fig. 5a), our method outperformed diffusion-based methods and achieved results nearly on par with GAN-based approaches. Moreover, our method recorded the highest recall metric of 0.56, significantly exceeding the recall of GAN methods, thereby underscoring its robustness in generating diverse and high-quality images.

Table 1: Class-conditional image generation on ImageNet $256 \times 256$ dataset.

| Model | FID↓ | Recall↑ | Params | #Iters × Bs | Epoch |
|---|---|---|---|---|---|
| Ours | **8.61** | **0.67** | 460M | 936K × 704 | 510 |
| Ours-G | **2.11** | **0.59** | 460M | 936K × 704 | 510 |
| SSM-based | | | | | |
| DIFFUSSM-XL [65] | 9.07 | 0.64 | 673M | 2578K× 256 | 515 |
| DIFFUSSM-XL-G | 2.28 | 0.56 | 673M | 2578K × 256 | 515 |
| UNet-based | | | | | |
| LDM-4 [51] | 10.56 | 0.62 | 400M | 178K × 1200 | 200 |
| LDM-4-G | 3.60 | 0.48 | 400M | 178K × 1200 | 200 |
| Transformer-based | | | | | |
| DiT-L/2 [48] | 23.33 | - | 458M | 400K × 256 | 80 |
| DiT-XL/2 | 9.62 | 0.67 | 675M | 7000K × 256 | 1.4K |
| DiT-XL/2-G | 2.27 | 0.57 | 675M | 7000K × 256 | 1.4K |
| SiT-XL/2 [44] | 9.40 | - | 675M | 7000K × 256 | 1.4K |
| SiT-XL/2-G | 2.15 | 0.59 | 675M | 7000K × 256 | 1.4K |
| GAN model | | | | | |
| BigGan-deep [4] | 6.95 | 0.28 | 160M | - | |
| StyleGAN-XL [54] | 2.30 | 0.53 | 166M | 25000K × 256 | 4K |

| | FID↓ | Recall↑ | Params | GFLOPs |
|---|---|---|---|---|
| Baseline | 6.19 | 0.46 | 413M | 51.65 |
| + Conditional Mamba | 5.27 | 0.49 | 446M | 51.69 |
| + Wavelet Mamba (w/ concat) | 5.87 | 0.47 | 394M | 56.54 |
| + Cross-Attention fusion layer | 4.92 | 0.50 | 436M | 62.42 |
| + Shared transformer block | 4.65 | 0.52 | 459M | 84.49 |

(a) Network components.

| #Lv | FID↓ | Recall↑ | GFLOPs |
|---|---|---|---|
| 1 | 5.09 | 0.50 | 84.48 |
| 2 | 4.65 | 0.52 | 84.49 |
| 3 | 5.23 | 0.49 | 84.50 |

(b) Wavelet levels.

| | FID↓ | Recall↑ | Params | GFLOPs |
|---|---|---|---|---|
| Linear | 6.00 | 0.47 | 411M | 60.83 |
| Attention | 5.05 | 0.50 | 461M | 73.71 |
| CAFL (swap q) | 4.92 | 0.50 | 436M | 67.27 |
| CAFL (swap k) | 5.45 | 0.48 | 436M | 67.27 |

(c) Fusion layers. CAFL means Cross-Attention Fusion Layer.

| | FID↓ | Recall↑ | Params | GFLOPs |
|---|---|---|---|---|
| DCT | 5.53 | 0.50 | 436M | 67.33 |
| EinFFT | 5.63 | 0.48 | 371M | 66.96 |
| Wavelet | 4.92 | 0.50 | 436M | 62.42 |

(d) Frequency types.

| Order | FID↓ | Recall↑ | iters/s ↑ |
|---|---|---|---|
| **Conditional Mamba Only** | | | |
| Bi | 6.39 | 0.44 | 2.06 |
| Sweep-4 | 5.27 | 0.49 | 2.06 |
| Sweep-8 | 5.53 | 0.48 | 1.97 |
| Zigzag-8 | 6.17 | 0.46 | 1.97 |
| Jpeg-8 | 6.26 | 0.45 | 1.97 |
| Window | 10.88 | 0.36 | 2.05 |
| **Spatial-frequency Mamba** | | | |
| Sweep-4 ǀ Sweep-4 | 5.41 | 0.49 | 1.54 |
| Sweep-4 ǀ Window | 4.92 | 0.50 | 1.54 |

(e) Scanning orders.

| | FID↓ | Recall↑ | Params | GFLOPs |
|---|---|---|---|---|
| **Conditional Mamba Only** | | | | |
| GS | 5.40 | 0.49 | 469M | 78.30 |
| **Spatial-Frequency Mamba** | | | | |
| Idp | 5.08 | 0.49 | 397M | 63.37 |
| GS | 4.65 | 0.52 | 459M | 84.49 |

(f) Transformer layers. GS stands for globally shared. Idp stands for Independent.

Table 2: Ablation studies on CelebA-HQ $256 \times 256$ dataset at epoch 250.

On the ImageNet1k 256 dataset, our methodology attains a formidable FID of 2.26 using a guidance scale of 1.4, surpassing the DiT models across comparable and larger model configurations, such as DiT-XL/2. This performance superiority extends to other benchmarks, including the SSM-based DIFFUSSM-XL-G model. Although our model yields results similar to those of the SiT model, our model is approximately 30% smaller in size.

## 4.2 Training convergence

As reported in Tables 1, 4, 5a, our method requires less training epochs/iterations to reach the optimal performance compared to the baseline approach, implying a strong and fast convergence. To better illustrate the training convergence comparison, we illustrate in Fig. 4d the performance of different diffusion-based models over training epochs regarding the FID-10K on CelebA-HQ 256. Notably, our proposed method demonstrates a superior convergence speed, rapidly decreasing FID score and stabilizing at a significantly lower value than the other methods like LFM[24], LDM[51], and DiT[48]. This rapid descent is particularly evident within the first 150 epochs, after which our method maintains a low FID score and still with sight on decrement, suggesting a high-quality image generation capability. Compared to the learning curves of other methods, our method exhibits a more stable trajectory without significant oscillations between training epochs.

## 4.3 Ablation of network design

In this section, we ablate the design choices for our network, using experiments on the CelebA-HQ 256 dataset. For the starting baseline, we adopt the same training settings from Zigma[25]. We choose sweep-4 with interleave scanning order [33] by default. In Fig. 2a, with our proposed conditional Mamba, the FID score is improved from 6.19 to 5.27, and the same trend is observed for recall. Meanwhile, adding Wavelet Mamba followed by a simple concatenation layer to combine spatial and wavelet features results in a worse score of 5.87 due to the weak alignment between these features. This demonstrates that our proposed cross-attention fusion layer is crucial for performance improvement, fully leveraging wavelet components to achieve a score boosted to 4.92. The performance is further enhanced to 4.66 by incorporating the weight-shared transformer block.

**Design choices of fusion layer.** In Fig. 2c, we report metrics for different fusion layers, ranging from simple linear projection to cross-attention layers. As shown, our proposed cross-attention fusion

with swapped query achieves the best results while requiring fewer parameters and GFLOPs than the attention option, with only a marginal increase in computation compared to the linear option.

**Number of wavelet levels.** As shown in Fig. 2b, two wavelet levels provide the best performance on the CelebA-HQ 256 dataset. We argue that the choice of wavelet levels should be based on the input resolution. An input image of size $256 \times 256$, for instance, is encoded to a compact latent of size $32 \times 32$, which is further patchified by 2 to the small size of $16 \times 16$. Hence, applying 3 wavelet levels results in extremely small wavelet subbands of size $2 \times 2$, leading to reduced performance.

**Alternative frequency transform.** Apart from wavelet transform, we also consider different types of frequency techniques like DCT and Fourier Transform (Fig. 2d). For DCT, we propose a multi-order JPEG scanning strategy (illustrated in Fig. 8), based on JPEG compression [63] instead of the window scanning. For Fourier Features, we directly adopt the EinFFT block from SiMBA [47]. In either case, the performance drops compared with the default choice of wavelet.

**Transformer layer.** In Fig. 2f, we assess the advantage of the transformer layer for both Conditional Mamba and Spatial-Frequency Mamba. As shown, the globally-shared transformer further boosts the performance of our Spatial-Frequency Mamba. In contrast, applying this layer solely to spatial Mamba increases the FID score by 0.13, highlighting the essence of our Spatial-Frequency Mamba in conjunction with the transformer layer. Meanwhile, replacing this shared layer with independent transformers results in a decline of 0.43 in FID and 0.03 in Recall.

### 4.4 Ablation of scanning orders

In Fig. 2e, we compare different scanning orders of Mamba. We keep it simple by using only Mamba block for all experiments without the globally-shared transformer and fusion layer. In Spatial-frequency Mamba, we use "Sweep-4" scanning for spatial features by default and only test other scanning methods for wavelet features. Overall, Sweep-4 performs best when combined with our proposed Conditional Mamba module. We also show that scanning with many-way orders like Sweep-8, Zigzag-8, and Jpeg-8 is not guaranteed to yield better performance than Sweep-4 scanning. In spatial-frequency Mamba, it is demonstrated that our proposed window scanning for wavelet Mamba provides a better outcome than conventional sweep-4 scanning.

## 5 Conclusion

Our paper introduces a novel, promising architecture that seamlessly integrates spatial and frequency features into Mamba process. By leveraging wavelet transform within the Mamba framework, our method enhances local structure awareness and ensures efficient spatial and frequency information fusion. This dual-focus strategy improves the detail and quality of generated images and accelerates training convergence. Our comprehensive experiments demonstrate that DiMSUM consistently outperforms state-of-the-art models of comparable size across multiple benchmarks, achieving lower FID scores and higher recall metrics, highlighting its ability to produce diverse and high-fidelity images. The proposed cross-attention fusion layer and globally shared transformer block also contribute to the model's robustness and scalability. Considering the promising results, we anticipate that future research in related domains, such as text-to-image synthesis, will adapt our backbone architecture and achieve comparable improvements in performance.

**Social impact and limitation.** We believe that our proposed network advances the architectural design of state-space models for image generation. This model can be extended to various tasks, such as large-scale text-to-image generation and multimodal diffusion. While there is a risk that our architecture could be misused for malicious purposes, posing a social security challenge, we are confident that this risk can be mitigated with the recent development of security-related research. Hence, the positives can outweigh the negatives, rendering the concern minor.

While our method outperforms other diffusion baselines in generation quality and training convergence, we acknowledge areas for improvement. These include designing a multiscale architecture and addressing manually defined scanning orders. Another advanced technique, such as masking training regularization [14, 72], is orthogonal to our approach and could lead to further improvements.

## Acknowledgements

Research partially funded by research grants to Prof. Dimitris Metaxas from NSF: 2310966, 2235405, 2212301, 2003874, 1951890, AFOSR 23RT0630, and NIH 2R01HL127661.

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

Table 3: Scaling DiMSUM's parameters on LSUN Church.

| Model | FID↓ | Epochs | Params |
|---|---|---|---|
| DiMSUM-L/2 | 3.76 | 395 | 460M |
| DiMSUM-XL/2 | **3.45** | 340 | 718M |
| DIFFUSSM | 3.94 | - | 673M |
| StyleGAN | 4.21 | - | - |
| StyleGAN2 | 3.86 | 13K | - |

# A   Training details

We provide network configuration details in table 6b and hyperparams in table 6a. For the interpolent form of the forward process eq. (1), we opt to use a generalized VP form, which is proposed in SiT paper [44]. Particularly, this form defines $\alpha_t = \cos\left(\frac{1}{2}\pi t\right)$ and $\sigma_t = \sin\left(\frac{1}{2}\pi t\right)$. The choice of transformer block, we adopt a DiT block [48] where we replace the original MLP layer with a Gated MLP layer, similar to SwitchTransformer [12].

Besides, we illustrate the sweep scanning strategy in Fig. 9 for better comprehension. We also visualize the Jpeg scanning orders in Fig. 8 inspired by the Jpeg compression algorithm [63] and the multi-direction scanning from ZigMa paper [25].

# B   Discussion

**Advantages of DiMSUM.** First, it's important to highlight that our method outperforms both DiT and SiT while requiring less than a third of the training iterations, achieving the best FID score of 2.11. Compared to other state-space diffusion models, our method outperforms DIFFUSSM-XL, considering a similar training duration. Notably, our method also uses a smaller network size of 460M parameters, compared to 675M of DiT while still demonstrating strong generation capacity and faster training convergence.

**Clarification of scalable term.** Given its current association with model parameter scaling in the LLM-dominated landscape, 'scalable' may be mistakenly interpreted as primarily referring to increasing the size of the architecture. However, in the broader context, machine learning/ deep learning algorithms' scalability may refer to their capacity to handle bigger datasets and computational resources while producing correct results in an acceptable period of time.

Our model aligns well with this definition of scalability. Our experiments demonstrated that:

- The model adapts to multiple datasets with minimal hyperparameter tuning (similar to DiT paper).
- It achieves competitive performance metrics compared to other methods.
- Inference speed is also faster, as shown in Table 4.

Our SoTA results are achieved with a parameter count comparable (or even smaller) to existing models. Refer to Table 4a, 5a and 1. This suggests substantial room for further enlargement of our model's parameters, which we anticipate will yield even greater improvements across various and bigger datasets (see Table 3).

**Why would using only wavelet scanning hurt model performance (as observed in Table 2a)?** We hypothesize that spatial and frequency signals are not aligned and require careful integration to leverage their information. Naively fusing these domains (e.g., by concatenation) can damage performance due to conflicting or misaligned information.

To address this challenge, we proposed a more sophisticated fusion method using Cross-Attention layers between these spaces. This approach enables the model to effectively combine information from both domains, leveraging their strengths while mitigating potential conflicts. Hence, this fusion technique can enhance the FID from 5.87 to 4.92 in Table 2c, contributing to the SoTA result of our method.

Table 4: Speed and GFLOPs comparison. Single-sample generation was performed, and all tests were conducted on an NVIDIA A100 40GB GPU.

| Method | Time | MEM | Params | GFlops |
|--------|------|-----|--------|--------|
| 256 (latent size: $32 \times 32$) | | | | |
| Ours-L/2 | 2.20s | 2.42G | 460M | 84.49 |
| DiT-L/2 | 3.80s | 2.30G | 458M | 80.74 |
| 512 (latent size: $64 \times 64$) | | | | |
| Ours-L/2 | 2.86s | 2.46G | 461M | 337.48 |
| DiT-L/2 | 4.78s | 2.34G | 459M | 361.14 |

## C  Speed Analysis

**Memory and GFLOPs.** The results, as shown in the table 4, reveal that DiMSUM-L/2's memory usage is slightly higher than its counterpart. This increase is expected, considering DiMSUM's slightly larger parameter count. Note that the parameter change after changing image size is mainly due to the PatchEmbed layer of the architecture, which both models have.

Regarding GFlops, we acknowledge that for $256 \times 256$ images, DiMSUM produces about 4% more GFlops than DiT. However, an interesting trend emerges when we examine 512x512 images: DiMSUM's GFlops scaling is actually slower than DiT's, proving the efficiency of our method for high-resolution image synthesis. Consequently, at this higher resolution, DiT's GFlops exceed DiMSUM's by approximately 7%. This observation further highlights the strength of Mamba in handling longer context length.

This observation aligns with the known quadratic complexity of transformers as sequence length increases. Our hybrid model mitigates this issue; the impact of attention blocks is reduced, while Mamba demonstrates its linear scaling complexity as the token count grows. This architectural choice allows DiMSUM to maintain efficiency at higher resolutions, offsetting the initial GFlops difference at 256 resolution.

With these two points, we emphasize upon the importance of our proposed architecture, rather than just the benefits given by the flow matching framework.

**Speed gain.** In table 4, DiMSUM-L/2 achieves 2.2 seconds latency compared to DIT-L/2 with 3.8 seconds though DiMSUM has larger GFLOPs. Notably, with resolution 512 (around 1024 tokens), the speed gap between our method and DiT becomes more significant and our architecture also has lower Gflops. This demonstrate the potential use of DiMSUM in larger benchmark like text-to-image which has larger resolution.

**Analysis**. Observing the Memory and GFLOPS in table table 4, it's true to claim DiMSUM-L/2 shows slower inference speed compared to its counterpart for 256x256 images using the same NFE (due to bigger GFLOPS), however, there are two crucial points to consider:

- Scaling Efficiency: When we increase the image size to 512x512, as evident from the Memory and GFLOPS table, our model actually requires fewer GFLOPS at this higher resolution, thanks to its slower latency scaling which we mentioned above. Consequently, for 512x512 images and larger, DiMSUM-L/2 would outperform its counterpart in speed given the same NFE.

- Adaptive Sampling Efficiency: We employ the dopri5 ODE adaptive solver for sampling from both models, similar to SiT [44] and Zigma [25]. This solver dynamically adjusts the NFE based on the initial noise and diffusion model characteristics, using the minimum NFE necessary to achieve optimal image quality. Notably, DiMSUM requires fewer NFE to meet the dopri5 stopping condition while still achieving a significantly better FID score than DiT. **We hypothesize that our proposed hybrid architecture converges to a better solution with less curvature, enabling high-fidelity image production with fewer NFEs**.

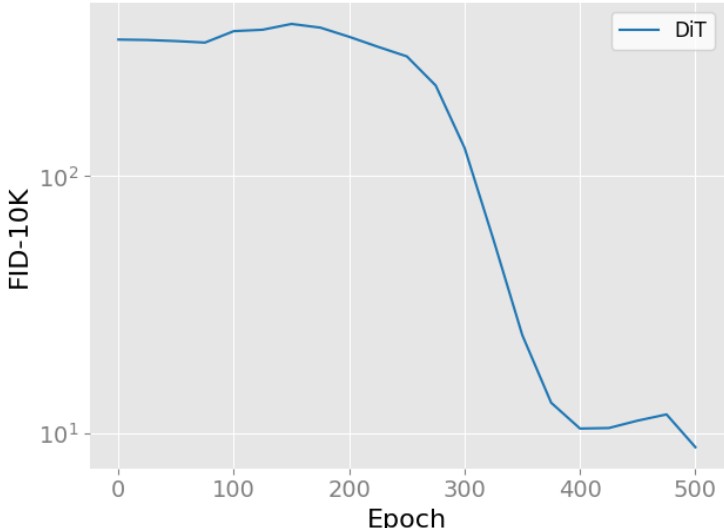

Figure 6: Training curve of DiT after training longer on CelebA, supporting Fig. 4d in the manuscript.

Table 5: Additional ablation of scanning orders involves using window scanning for wavelet blocks and examining the sensitivity of DiMSUM to different scanning orders for spatial blocks. For simplicity, no shared-transformer block is used.

| Order | freq + spatial | spatial |
|---|---|---|
| Bi | 5.47 | 6.39 |
| Jpeg | 5.74 | 6.26 |
| sweep-8 | 5.23 | 5.53 |
| zigma-8 | 5.71 | 6.26 |
| sweep-4 | **4.92** | 5.28 |

With these two points, we emphasize upon the importance of our proposed architecture, rather than just the benefits given by the flow matching framework.

## D    More quantitative results

**Full training curve of DiT.** The plot Fig. 4d intentionally stops at epoch 300, demonstrating the model's capability to converge faster than other methods. DiT does perform well on CelebA-HQ but takes more than 500. Here, we further provide a complete training curve of DiT in Fig. 6. It's noted that DiT and LFM in Fig. 4d use the same DiT-L/2 architecture. While DiT uses diffusion loss, LFM uses flow matching. LFM converge faster than DiT. However, our model with flow matching loss, demonstrates even faster convergence than LFM, indicating our architecture enhances convergence rate.

**Additional ablation of scanning orders.** To substantiate our claims regarding the efficiency of frequency information, we conducted a comprehensive ablation study. This ablation utilized four scanning orders: (1) bidirection, (2) jpeg-8, (3) sweep-8, and (4) zigzag-8. We trained models on CelebA-HQ at 256x256 resolution for 250 epochs, comparing performance when applying these scanning strategies to: a) spatial domain only or b) both spatial and frequency domains.

Table 5 shows that integrating frequency domain information across all four scanning strategies led to significant performance improvements. This consistent enhancement across various scanning methods provides strong evidence for the effectiveness of our approach in leveraging frequency information.

**Varying NFE.** We plot the FID-10K scores of various NFE used for evaluation in Fig. 7. This shows that increasing NFE beyond 250 leads to minimal or no improvement in the FID scores. This behavior

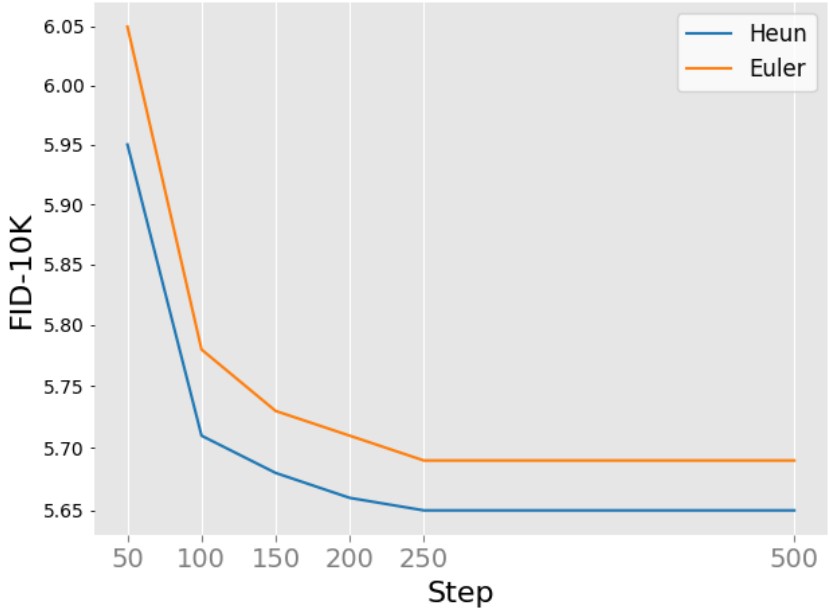

Figure 7: FID-10K of varying NFE on CelebA 256.

Table 6: Hyper-parameters and network config of our DiMSUM network.

(a) Hyper-parameters

|  | CelebA 256 & 512 | Church | ImageNet |
|---|---|---|---|
| Learning rate | 1e-4 | 5e-5 | 1e-4 |
| $\beta_1, \beta_2$ | 0.9, 0.999 | 0.9, 0.999 | 0.9, 0.999 |
| Batch size | 64 & 32 | 128 | 704 |
| Droppath | 0.1 | 0.2 | 0.1 |
| Max-grad-norm | 2 | 2 | 1 |
| Label-dropout | 0. | 0. | 0.15 |
| Epochs | 250 & 165 | 395 | 320 |
| GPUs (A100) | 2 & 4 | 4 | 8 |
| Train days | 0.89 & 3.2 | 3.42 | 12 |

(b) Network config

| Config | Value |
|---|---|
| Depth | 16 |
| Hidden size | 1024 |
| Patch size | 2 |
| Use learnable absolute positional embedding | True |
| Attention every k layer | 4 |

is consistent with the observation of the flow-matching in Fig. 7 of FM paper [36], which has been shown to require fewer NFEs compared to other SDE-based methods.

# E More qualitative examples

We present our uncurated generated samples of CelebA-HQ 256 in Fig. 10, CelebA-HQ 512 in Fig. 12, LSUN Church in Fig. 11, and ImageNet in Fig. 17, 18, 13, 14, 19, 20.

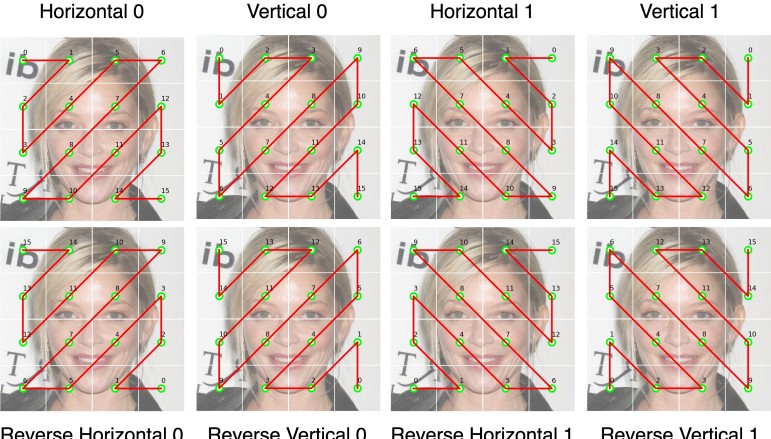

Figure 8: Illustration of Jpeg-8 scanning orders.

Figure 9: Illustration of Sweep scanning orders.


Figure 10: Uncurated generated samples of CelebA-HQ 256.

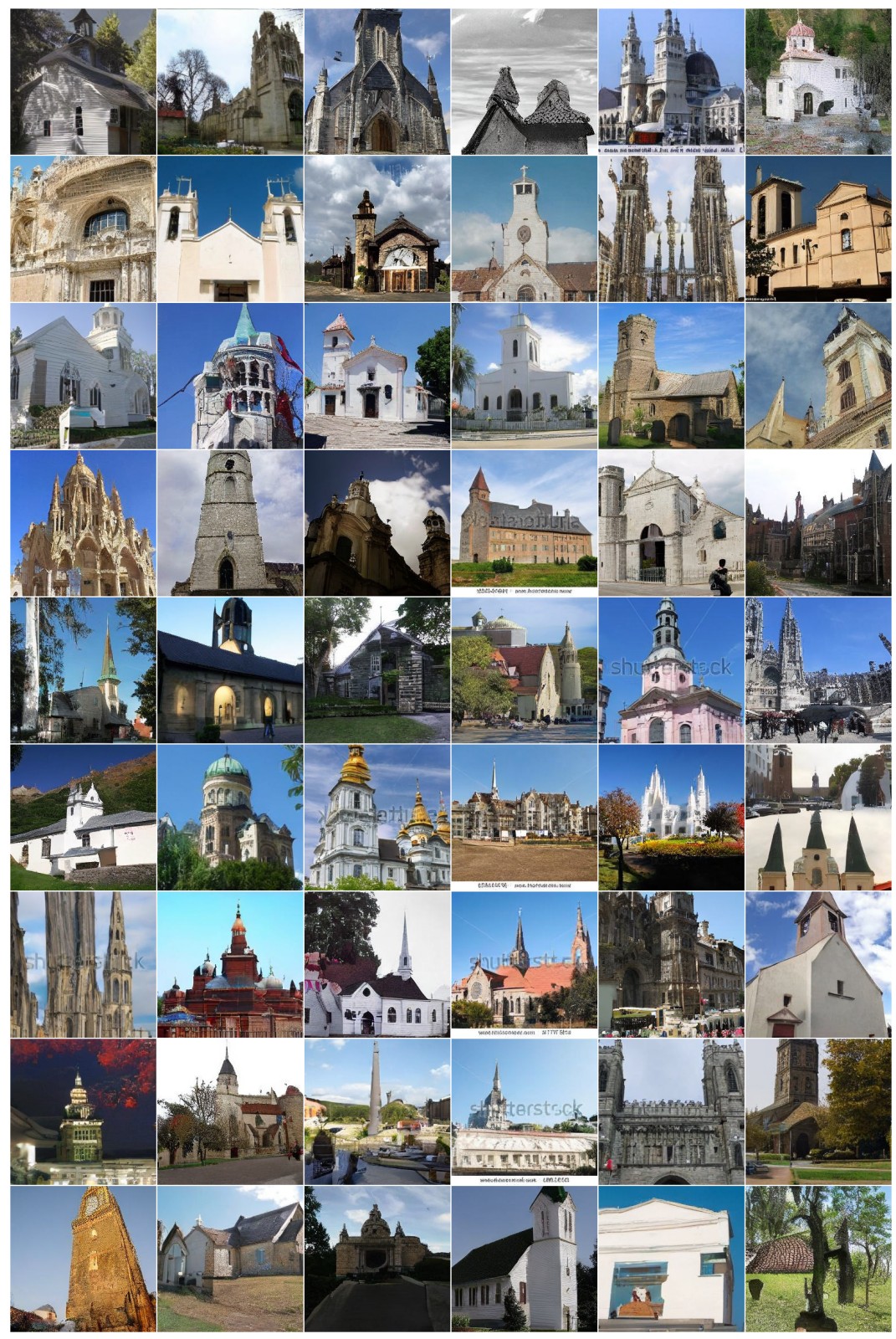

Figure 11: Uncurated generated samples of LSUN Church 256.

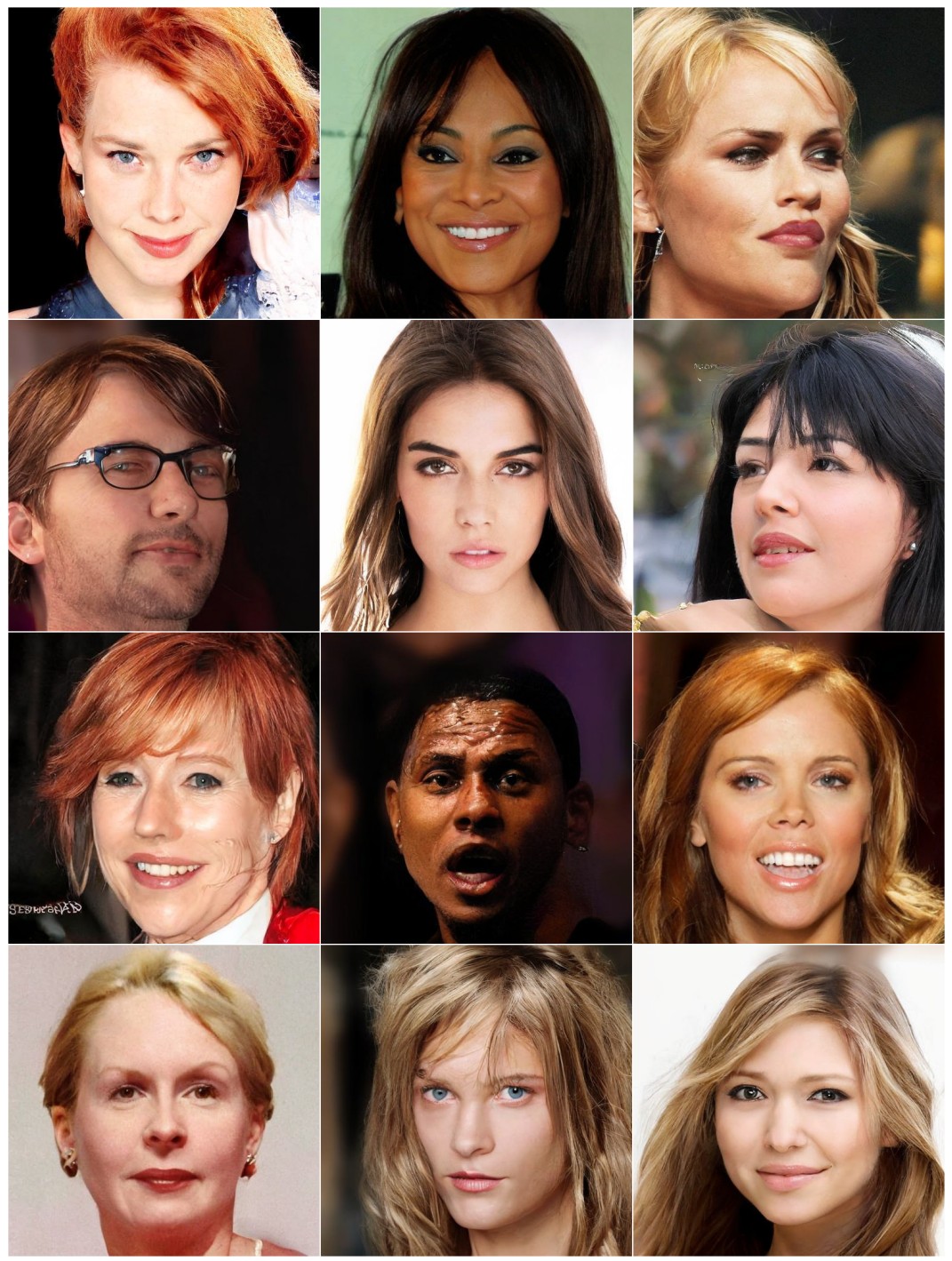

Figure 12: Uncurated generated samples of CelebA-HQ 512.

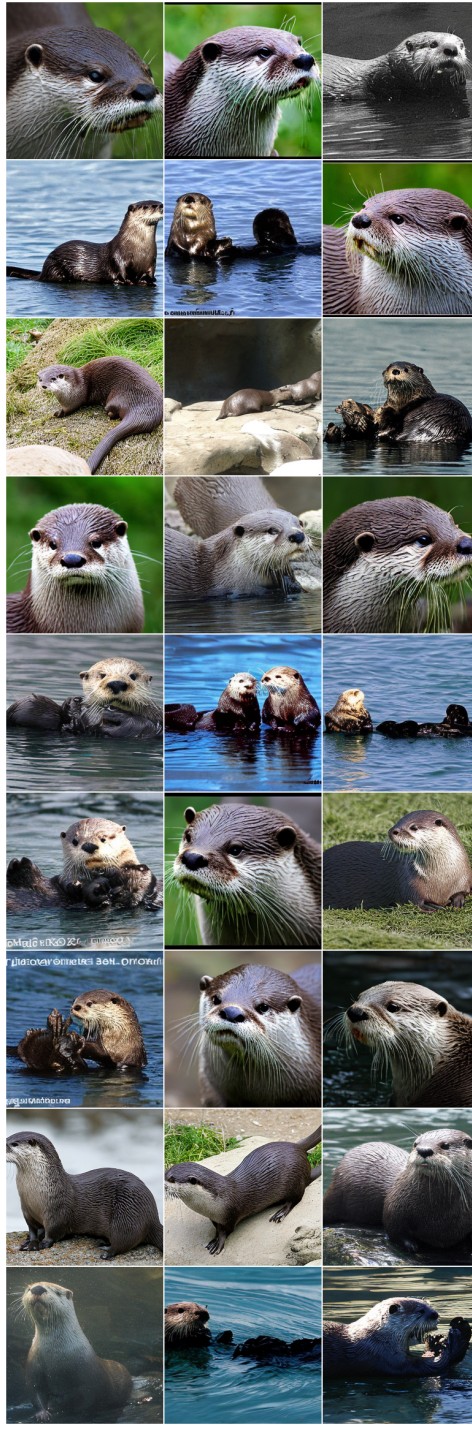

Figure 13: Uncurated generated samples of ImageNet 256 class 360 (otter) with cfg=4.0.

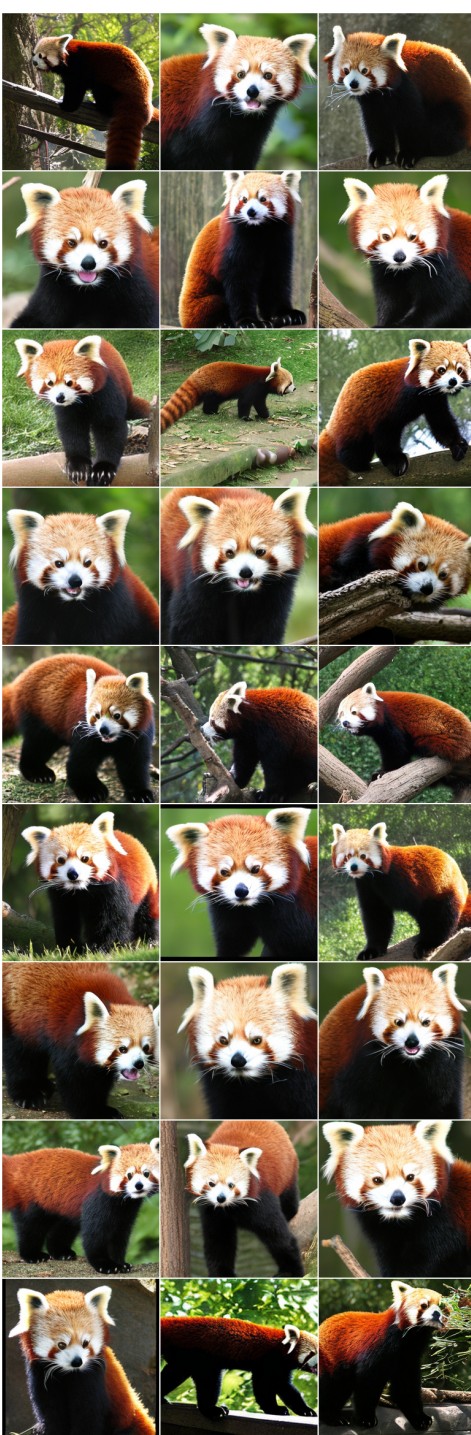

Figure 14: Uncurated generated samples of ImageNet 256 class 387 (lesser panda, red panda, panda, bear cat, cat bear, Ailurus fulgens) with cfg=4.0.

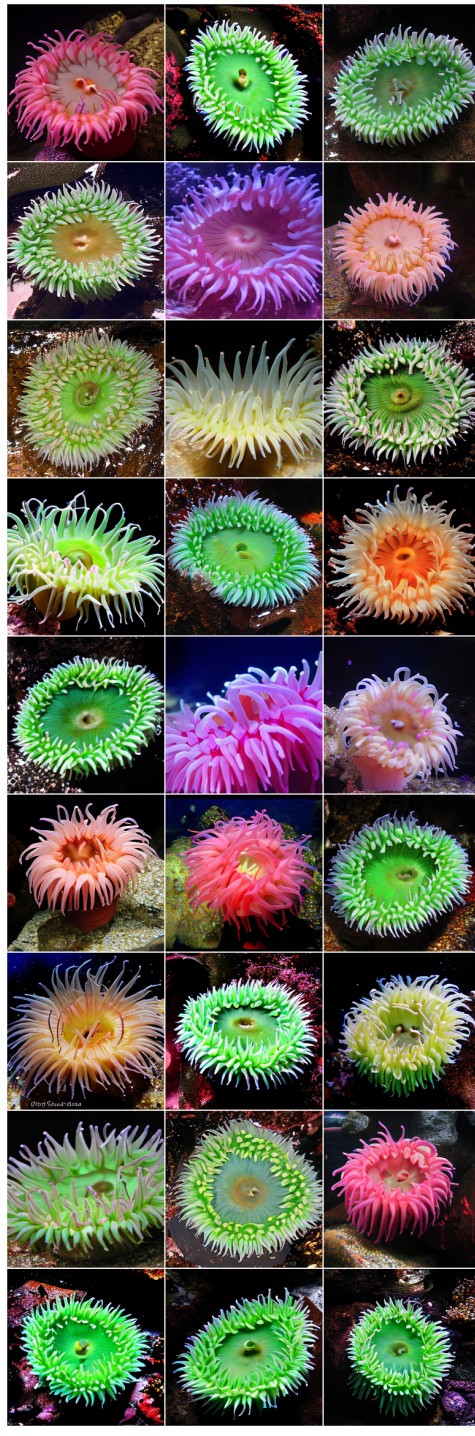

Figure 15: Uncurated generated samples of ImageNet 256 class 108 (sea anemone) with cfg=4.0.

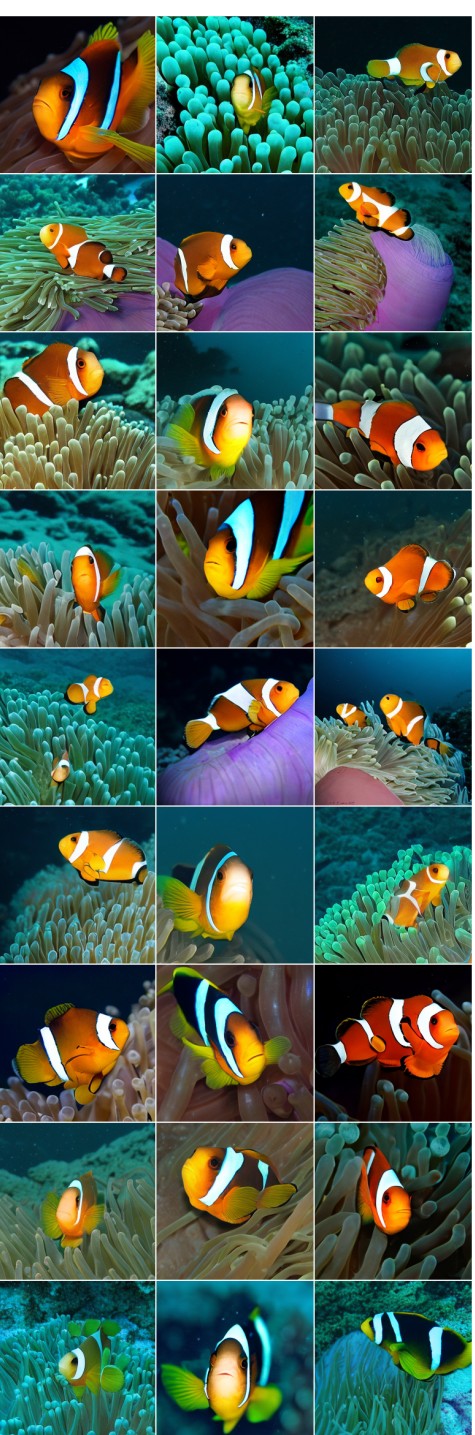

Figure 16: Uncurated generated samples of ImageNet 256 class 393 (anemone fish) with cfg=4.0.

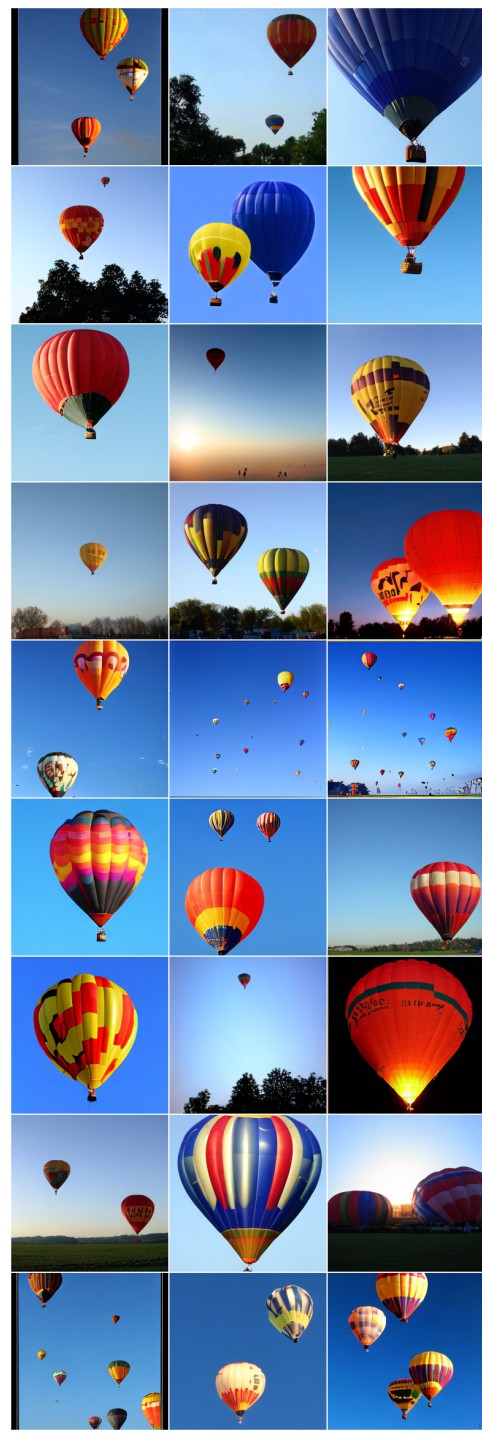

Figure 17: Uncurated generated samples of ImageNet 256 class 417 (balloon) with cfg=4.0.

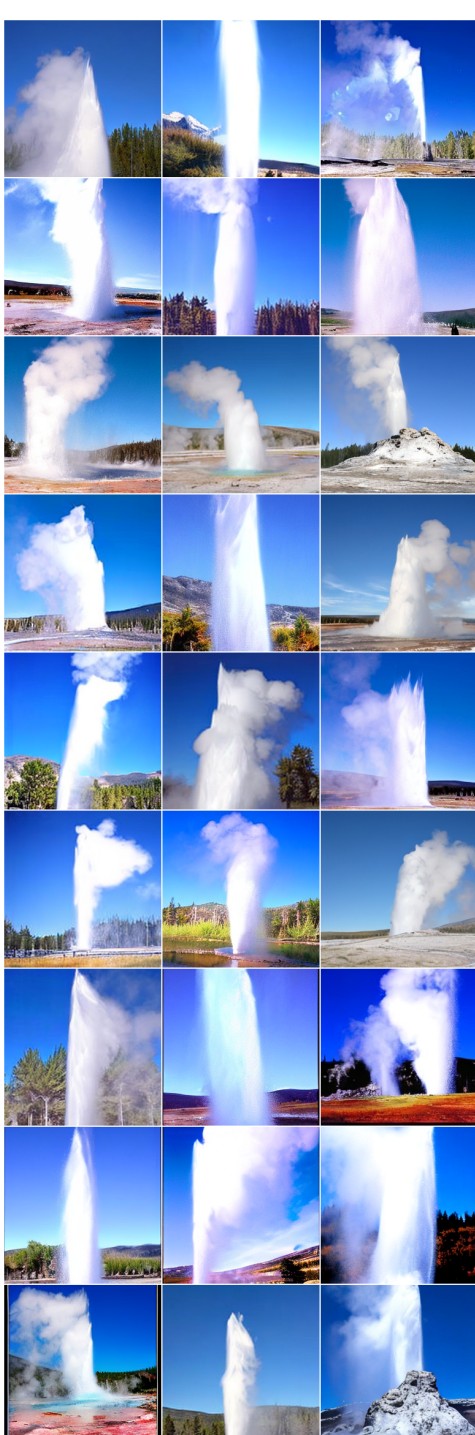

Figure 18: Uncurated generated samples of ImageNet 256 class 974 (geyser) with cfg=4.0.

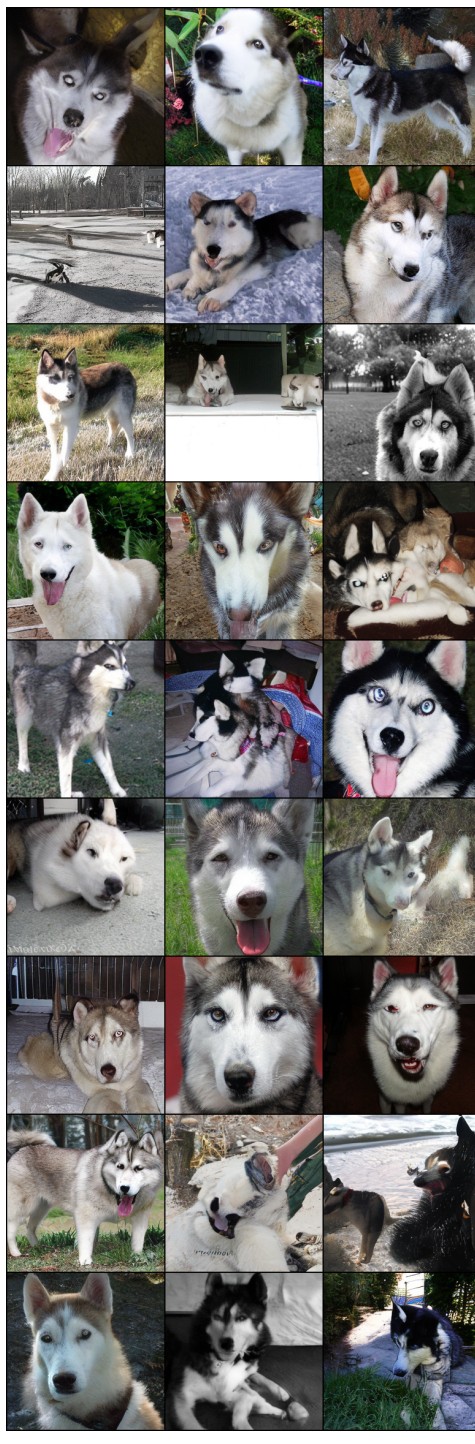

Figure 19: Uncurated generated samples of ImageNet 256 class 250 (Siberian husky) with cfg=1.5.

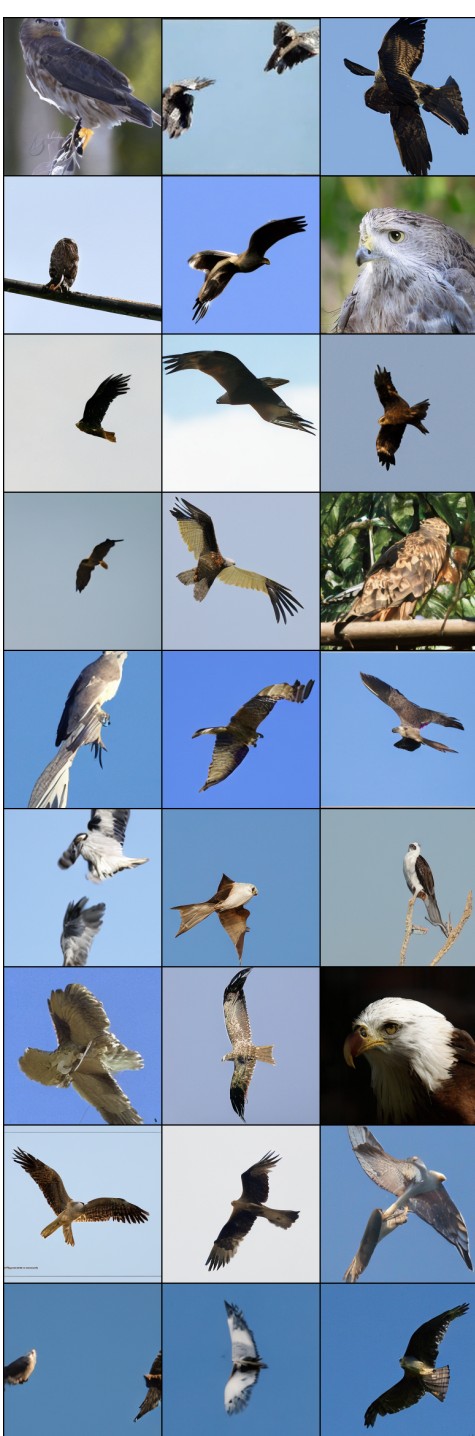

Figure 20: Uncurated generated samples of ImageNet 256 class 21 (bald eagle, American eagle, Haliaeetus leucocephalus) with cfg=1.5.

- It is fine to include aspirational goals as motivation as long as it is clear that these goals are not attained by the paper.

2. **Limitations**

Question: Does the paper discuss the limitations of the work performed by the authors?

Answer: [Yes]

Justification: We explicitly mention the limitation and suggest potential directions for future works in section 5.

Guidelines:

- The answer NA means that the paper has no limitation while the answer No means that the paper has limitations, but those are not discussed in the paper.
- The authors are encouraged to create a separate "Limitations" section in their paper.
- The paper should point out any strong assumptions and how robust the results are to violations of these assumptions (e.g., independence assumptions, noiseless settings, model well-specification, asymptotic approximations only holding locally). The authors should reflect on how these assumptions might be violated in practice and what the implications would be.
- The authors should reflect on the scope of the claims made, e.g., if the approach was only tested on a few datasets or with a few runs. In general, empirical results often depend on implicit assumptions, which should be articulated.
- The authors should reflect on the factors that influence the performance of the approach. For example, a facial recognition algorithm may perform poorly when image resolution is low or images are taken in low lighting. Or a speech-to-text system might not be used reliably to provide closed captions for online lectures because it fails to handle technical jargon.
- The authors should discuss the computational efficiency of the proposed algorithms and how they scale with dataset size.
- If applicable, the authors should discuss possible limitations of their approach to address problems of privacy and fairness.
- While the authors might fear that complete honesty about limitations might be used by reviewers as grounds for rejection, a worse outcome might be that reviewers discover limitations that aren't acknowledged in the paper. The authors should use their best judgment and recognize that individual actions in favor of transparency play an important role in developing norms that preserve the integrity of the community. Reviewers will be specifically instructed to not penalize honesty concerning limitations.

3. **Theory Assumptions and Proofs**

Question: For each theoretical result, does the paper provide the full set of assumptions and a complete (and correct) proof?

Answer: [NA]

Justification: N/A, because our work focuses solely on empirical evaluation to measure the effectiveness of the proposed architecture.

Guidelines:

- The answer NA means that the paper does not include theoretical results.
- All the theorems, formulas, and proofs in the paper should be numbered and cross-referenced.
- All assumptions should be clearly stated or referenced in the statement of any theorems.
- The proofs can either appear in the main paper or the supplemental material, but if they appear in the supplemental material, the authors are encouraged to provide a short proof sketch to provide intuition.
- Inversely, any informal proof provided in the core of the paper should be complemented by formal proofs provided in appendix or supplemental material.
- Theorems and Lemmas that the proof relies upon should be properly referenced.

4. **Experimental Result Reproducibility**

Question: Does the paper fully disclose all the information needed to reproduce the main experimental results of the paper to the extent that it affects the main claims and/or conclusions of the paper (regardless of whether the code and data are provided or not)?

Answer: [Yes]

Justification: Yes, this information can be found in appendix A and section 4.

Guidelines:

- The answer NA means that the paper does not include experiments.
- If the paper includes experiments, a No answer to this question will not be perceived well by the reviewers: Making the paper reproducible is important, regardless of whether the code and data are provided or not.
- If the contribution is a dataset and/or model, the authors should describe the steps taken to make their results reproducible or verifiable.
- Depending on the contribution, reproducibility can be accomplished in various ways. For example, if the contribution is a novel architecture, describing the architecture fully might suffice, or if the contribution is a specific model and empirical evaluation, it may be necessary to either make it possible for others to replicate the model with the same dataset, or provide access to the model. In general. releasing code and data is often one good way to accomplish this, but reproducibility can also be provided via detailed instructions for how to replicate the results, access to a hosted model (e.g., in the case of a large language model), releasing of a model checkpoint, or other means that are appropriate to the research performed.
- While NeurIPS does not require releasing code, the conference does require all submissions to provide some reasonable avenue for reproducibility, which may depend on the nature of the contribution. For example
  (a) If the contribution is primarily a new algorithm, the paper should make it clear how to reproduce that algorithm.
  (b) If the contribution is primarily a new model architecture, the paper should describe the architecture clearly and fully.
  (c) If the contribution is a new model (e.g., a large language model), then there should either be a way to access this model for reproducing the results or a way to reproduce the model (e.g., with an open-source dataset or instructions for how to construct the dataset).
  (d) We recognize that reproducibility may be tricky in some cases, in which case authors are welcome to describe the particular way they provide for reproducibility. In the case of closed-source models, it may be that access to the model is limited in some way (e.g., to registered users), but it should be possible for other researchers to have some path to reproducing or verifying the results.

5. **Open access to data and code**

Question: Does the paper provide open access to the data and code, with sufficient instructions to faithfully reproduce the main experimental results, as described in supplemental material?

Answer: [Yes]

Justification: Yes, we will release our code and pretrained models on github.

Guidelines:

- The answer NA means that paper does not include experiments requiring code.
- Please see the NeurIPS code and data submission guidelines (`https://nips.cc/public/guides/CodeSubmissionPolicy`) for more details.
- While we encourage the release of code and data, we understand that this might not be possible, so "No" is an acceptable answer. Papers cannot be rejected simply for not including code, unless this is central to the contribution (e.g., for a new open-source benchmark).
- The instructions should contain the exact command and environment needed to run to reproduce the results. See the NeurIPS code and data submission guidelines (`https://nips.cc/public/guides/CodeSubmissionPolicy`) for more details.

- The authors should provide instructions on data access and preparation, including how to access the raw data, preprocessed data, intermediate data, and generated data, etc.
- The authors should provide scripts to reproduce all experimental results for the new proposed method and baselines. If only a subset of experiments are reproducible, they should state which ones are omitted from the script and why.
- At submission time, to preserve anonymity, the authors should release anonymized versions (if applicable).
- Providing as much information as possible in supplemental material (appended to the paper) is recommended, but including URLs to data and code is permitted.

6. **Experimental Setting/Details**

Question: Does the paper specify all the training and test details (e.g., data splits, hyper-parameters, how they were chosen, type of optimizer, etc.) necessary to understand the results?

Answer: [Yes]

Justification: Yes, we provided them in Implementation part of section 4 and Training details in appendix A.

Guidelines:

- The answer NA means that the paper does not include experiments.
- The experimental setting should be presented in the core of the paper to a level of detail that is necessary to appreciate the results and make sense of them.
- The full details can be provided either with the code, in appendix, or as supplemental material.

7. **Experiment Statistical Significance**

Question: Does the paper report error bars suitably and correctly defined or other appropriate information about the statistical significance of the experiments?

Answer: [No]

Justification: No, because the evaluation process takes hours on a GPU and is very expensive to compute with multiple seeds. Importantly, we conducted evaluation for several benchmarks in section 4 so running with different seeds are not feasible in our scope.

Guidelines:

- The answer NA means that the paper does not include experiments.
- The authors should answer "Yes" if the results are accompanied by error bars, confidence intervals, or statistical significance tests, at least for the experiments that support the main claims of the paper.
- The factors of variability that the error bars are capturing should be clearly stated (for example, train/test split, initialization, random drawing of some parameter, or overall run with given experimental conditions).
- The method for calculating the error bars should be explained (closed form formula, call to a library function, bootstrap, etc.)
- The assumptions made should be given (e.g., Normally distributed errors).
- It should be clear whether the error bar is the standard deviation or the standard error of the mean.
- It is OK to report 1-sigma error bars, but one should state it. The authors should preferably report a 2-sigma error bar than state that they have a 96% CI, if the hypothesis of Normality of errors is not verified.
- For asymmetric distributions, the authors should be careful not to show in tables or figures symmetric error bars that would yield results that are out of range (e.g. negative error rates).
- If error bars are reported in tables or plots, The authors should explain in the text how they were calculated and reference the corresponding figures or tables in the text.

8. **Experiments Compute Resources**

Question: For each experiment, does the paper provide sufficient information on the computer resources (type of compute workers, memory, time of execution) needed to reproduce the experiments?

Answer: [Yes]

Justification: Yes, this is presented in appendix A.

Guidelines:

- The answer NA means that the paper does not include experiments.
- The paper should indicate the type of compute workers CPU or GPU, internal cluster, or cloud provider, including relevant memory and storage.
- The paper should provide the amount of compute required for each of the individual experimental runs as well as estimate the total compute.
- The paper should disclose whether the full research project required more compute than the experiments reported in the paper (e.g., preliminary or failed experiments that didn't make it into the paper).

9. **Code Of Ethics**

Question: Does the research conducted in the paper conform, in every respect, with the NeurIPS Code of Ethics `https://neurips.cc/public/EthicsGuidelines`?

Answer: [Yes]

Justification: Yes, we totally confirm to it.

Guidelines:

- The answer NA means that the authors have not reviewed the NeurIPS Code of Ethics.
- If the authors answer No, they should explain the special circumstances that require a deviation from the Code of Ethics.
- The authors should make sure to preserve anonymity (e.g., if there is a special consideration due to laws or regulations in their jurisdiction).

10. **Broader Impacts**

Question: Does the paper discuss both potential positive societal impacts and negative societal impacts of the work performed?

Answer: [Yes]

Justification: We discussed the societal imparts in section 5.

Guidelines:

- The answer NA means that there is no societal impact of the work performed.
- If the authors answer NA or No, they should explain why their work has no societal impact or why the paper does not address societal impact.
- Examples of negative societal impacts include potential malicious or unintended uses (e.g., disinformation, generating fake profiles, surveillance), fairness considerations (e.g., deployment of technologies that could make decisions that unfairly impact specific groups), privacy considerations, and security considerations.
- The conference expects that many papers will be foundational research and not tied to particular applications, let alone deployments. However, if there is a direct path to any negative applications, the authors should point it out. For example, it is legitimate to point out that an improvement in the quality of generative models could be used to generate deepfakes for disinformation. On the other hand, it is not needed to point out that a generic algorithm for optimizing neural networks could enable people to train models that generate Deepfakes faster.
- The authors should consider possible harms that could arise when the technology is being used as intended and functioning correctly, harms that could arise when the technology is being used as intended but gives incorrect results, and harms following from (intentional or unintentional) misuse of the technology.
- If there are negative societal impacts, the authors could also discuss possible mitigation strategies (e.g., gated release of models, providing defenses in addition to attacks, mechanisms for monitoring misuse, mechanisms to monitor how a system learns from feedback over time, improving the efficiency and accessibility of ML).

11. **Safeguards**

Question: Does the paper describe safeguards that have been put in place for responsible release of data or models that have a high risk for misuse (e.g., pretrained language models, image generators, or scraped datasets)?

Answer: [Yes]

Justification: Yes, our code will be released with Licences.

Guidelines:

- The answer NA means that the paper poses no such risks.
- Released models that have a high risk for misuse or dual-use should be released with necessary safeguards to allow for controlled use of the model, for example by requiring that users adhere to usage guidelines or restrictions to access the model or implementing safety filters.
- Datasets that have been scraped from the Internet could pose safety risks. The authors should describe how they avoided releasing unsafe images.
- We recognize that providing effective safeguards is challenging, and many papers do not require this, but we encourage authors to take this into account and make a best faith effort.

12. **Licenses for existing assets**

Question: Are the creators or original owners of assets (e.g., code, data, models), used in the paper, properly credited and are the license and terms of use explicitly mentioned and properly respected?

Answer: [Yes]

Justification: Yes, we have properly cited the code, data and models used.

Guidelines:

- The answer NA means that the paper does not use existing assets.
- The authors should cite the original paper that produced the code package or dataset.
- The authors should state which version of the asset is used and, if possible, include a URL.
- The name of the license (e.g., CC-BY 4.0) should be included for each asset.
- For scraped data from a particular source (e.g., website), the copyright and terms of service of that source should be provided.
- If assets are released, the license, copyright information, and terms of use in the package should be provided. For popular datasets, `paperswithcode.com/datasets` has curated licenses for some datasets. Their licensing guide can help determine the license of a dataset.
- For existing datasets that are re-packaged, both the original license and the license of the derived asset (if it has changed) should be provided.
- If this information is not available online, the authors are encouraged to reach out to the asset's creators.

13. **New Assets**

Question: Are new assets introduced in the paper well documented and is the documentation provided alongside the assets?

Answer: [Yes]

Justification: Our released source code is accompanied by an instruction document to run. Details about implementation can be found in section section 4.

Guidelines:

- The answer NA means that the paper does not release new assets.
- Researchers should communicate the details of the dataset/code/model as part of their submissions via structured templates. This includes details about training, license, limitations, etc.

- The paper should discuss whether and how consent was obtained from people whose asset is used.
- At submission time, remember to anonymize your assets (if applicable). You can either create an anonymized URL or include an anonymized zip file.

14. **Crowdsourcing and Research with Human Subjects**

Question: For crowdsourcing experiments and research with human subjects, does the paper include the full text of instructions given to participants and screenshots, if applicable, as well as details about compensation (if any)?

Answer: [NA]

Justification: The paper does not involve crowdsourcing nor research with human subjects.

Guidelines:

- The answer NA means that the paper does not involve crowdsourcing nor research with human subjects.
- Including this information in the supplemental material is fine, but if the main contribution of the paper involves human subjects, then as much detail as possible should be included in the main paper.
- According to the NeurIPS Code of Ethics, workers involved in data collection, curation, or other labor should be paid at least the minimum wage in the country of the data collector.

15. **Institutional Review Board (IRB) Approvals or Equivalent for Research with Human Subjects**

Question: Does the paper describe potential risks incurred by study participants, whether such risks were disclosed to the subjects, and whether Institutional Review Board (IRB) approvals (or an equivalent approval/review based on the requirements of your country or institution) were obtained?

Answer: [NA]

Justification: The paper does not involve crowdsourcing nor research with human subjects.

Guidelines:

- The answer NA means that the paper does not involve crowdsourcing nor research with human subjects.
- Depending on the country in which research is conducted, IRB approval (or equivalent) may be required for any human subjects research. If you obtained IRB approval, you should clearly state this in the paper.
- We recognize that the procedures for this may vary significantly between institutions and locations, and we expect authors to adhere to the NeurIPS Code of Ethics and the guidelines for their institution.
- For initial submissions, do not include any information that would break anonymity (if applicable), such as the institution conducting the review.

