# OpenReview forum: "DiMSUM: Diffusion Mamba - A Scalable and Unified Spatial-Frequency Method for Image Generation"
_NeurIPS.cc/2024/Conference — NeurIPS 2024 poster_

### Official Review · Reviewer_C5pc · 2024-06-29

**Soundness:** 3
**Presentation:** 2
**Contribution:** 2
**Rating:** 4
**Confidence:** 5

**Summary:**

This paper introduces a state-space architecture for diffusion models that enhances local feature detection in image generation by combining spatial and frequency information. Traditional state-space networks like Mamba struggle with visual data processing, but integrating wavelet transformation improves local structure awareness. The method fuses wavelet-transformed outputs with original Mamba outputs and incorporates a globally-shared transformer to capture global relationships. Experiments show this approach achieves faster training convergence and high-quality outputs, demonstrating state-of-the-art results.

**Strengths:**

+ The proposed method works on the frequency domain seems to be a good idea compared to existing mamba diffusion approaches.

+ The architecture looks like getting inspiration from DiT and transferring to the Mamba style may be interesting.

+ Performance on several datasets shows the promising method.

**Weaknesses:**

+ Training longer looks like getting an overfitting curve, an analysis needs to be done. This behavior is contradict to Transformer-based diffusion DiT, MDT, MaskDiT, etc.

+ The paper claims the issue with quadratic computation of transformers-based diffusion but there is no comparison between the proposed method and the competitor for both 256 and 512 resolution.

+ The scalability is important for this architecture, but it is missing, which would weaken the paper.

+ On ImageNet, the crucial benchmark, the reported performance of the proposed method is not clearly better than DiT, and of course, cannot be comparable with the more advanced method transformer-based model such as MDT. This indicates that the proposed method seems still under-discovered, not better than the existing methods in the performance of image generation.

**Questions:**

Q1. How is performance if using the same sampling method as other competitors that are based on DDIM/DDPM, instead of ODE solver? This is to ensure a fair comparison.

Q2. How is the performance w.r.t more sampling steps? A curve is a good demonstration.

**Limitations:**

Yes, they discussed

---

> ### Author Rebuttal · Authors · 2024-08-07
>
> 1. Training longer looks like getting an overfitting curve, an analysis needs to be done. This behavior is contradict to Transformer-based diffusion DiT, MDT, MaskDiT, etc.
>
> Thank you for pointing out, we confirm that our method does have overfitting. After converging to the best FID10K score of 5.49 at epoch 250, the model begins to exhibit an overfitting pattern. We also observe the same overfitting problem for transformer-based diffusion models like DiT and MDT. Figure 3 in the rebuttal file demonstrates that overfitting does occur in transformer-based models, albeit at a later stage, starting from epoch 800 onward, due to their slower convergence rate compared to our model.
>
> We hypothesize that overfitting is not solely determined by model architecture but by various factors. For instance, the Improved DDPM [2] paper showed that with identical architecture in Appendix F, changing the noise scheduling from linear to cosine could lead to overfitting in same UNET architecture. Based on our empirical experiments above, we think that the overfitting effect is not only attributable to our architecture but rather a broader phenomenon in diffusion models. For MDT, overfitting emerges much later, around epoch 1100, due to its masking strategy, which could serves as a form of regularization to delay the onset of overfitting.
>
> 2. The paper claims the issue with quadratic computation of transformers-based diffusion but there is no comparison between the proposed method and the competitor for both 256 and 512 resolution.
>
> DIMSUM-L/2 achieves 2.2 seconds latency compared to DIT-L/2 with 3.8 seconds (see table 2 in rebuttal file).
>
> 3. The scalability is important for this architecture, but it is missing, which would weaken the paper.
>
> For this question, we provide our answers in the global response.
>
> 4. On ImageNet, the crucial benchmark, the reported performance of the proposed method is not clearly better than DiT, and of course, cannot be comparable with the more advanced method transformer-based model such as MDT. This indicates that the proposed method seems still under-discovered, not better than the existing methods in the performance of image generation.
>
> To answer this question, we break it down into two parts: (1) not clearly outperforming DiT and (2) underperforming compared to MDT.
>
> First, it is important to highlight that our method achieves competitive performance with DiT and DIFFUSSM-XL, requiring significantly fewer training epochs—specifically, 4x fewer than DiT and 2x fewer than DIFFUSSM-XL. Notably, our method also uses a smaller network size of 460M parameters, compared to 675M of DiT while still demonstrating strong generation capacity and faster training convergence. In the updated ImageNet table provided in the PDF, we show that our method can further surpass the performance of other diffusion baselines when trained for a similar training duration as DIFFUSSM-XL. Notably, our training iterations are still less than a third of those required by DiT and SiT, yielding the best FID score of 2.11.
>
> Though our method does not outperforms the results of MDT yet, we believe that MDT is orthogonal to our proposed architecture, where a masking scheme is introduced to enhance further the contextual learning ability of diffusion models, including our method. It is an interesting topic to combine with our network for boosting model performance as mentioned in the limitation of global response.
>
> 5. Q1. How is performance if using the same sampling method as other competitors that are based on DDIM/DDPM, instead of ODE solver? This is to ensure a fair comparison.
>
> To clarify, Euler solver and DDIM are the same, except that Euler operates on continuous time intervals and removes the additional stochastic noise in each denoising step. In the manuscript, we followed SiT, Zigma, LFM works on using adaptive solver like "dopri5" for evaluation. In table 5 of the attached PDF, we show that FID scores of adaptive ODE solver and Euler solver (or DDIM sampling) are similar with small numerical difference, reconfirming the fairness of the benchmarking with DiT.
>
> 6. Q2. How is the performance w.r.t more sampling steps? A curve is a good demonstration.
>
> We plot the FID-10K scores of various NFE used for evaluation in Fig. of the attached pdf. This shows that increasing NFE beyond 250 leads to minimal or no improvement in the FID scores. This behavior is consistent with the observation of the flow-matching in Figure 7 of FM paper, which has been shown to require fewer NFEs compared to other SDE-based methods.
>
> [1] Lipman, Yaron, et al. "Flow matching for generative modeling." ICLR 2023.
>
> [2] Nichol, Alexander Quinn, and Prafulla Dhariwal. "Improved denoising diffusion probabilistic models." International conference on machine learning. PMLR, 2021.

---

> > ### Comment · Reviewer_C5pc · 2024-08-12
> >
> > Thanks for the rebuttal, it addressed some concerns, still remained some unclear points.
> >
> > 1) How is the latency comparison with 512x512?
> >
> > 2) Memory use is an important metric when claiming the efficiency of the proposed method (in Line 127), could authors provide Memory Use and Gflop besides a comparison with others for 256 and 512 resolution they showed in Tab. 2 rebuttal?
> >
> > 3) Could the authors explain some insights as to why their method is faster than DiT although it has more parameters than DiT?
> >
> > 4) In the proposed method, the input image is further processed with several handcrafted features of wavelet transforms and scanning operations, before patchifying and feeding into the model, which will also be considered to take more processing time to slow down the speed, are they accounted for in the inference speed when making a comparison for both 256 and 512 resolutions, and are they used the same NFE to compare?
> >
> > 5) In the paper, it mentioned transformers-based methods, including MDT, and MaskDiT, and the proposed method obviously cannot be comparable with these models, but the abstract claims it achieves state-of-the-art, which I believe is misleading and inappropriate.

---

> ### Author Response · Authors · 2024-08-13
>
> Thank you for your response! We provide answers below:
>
> Q1. 512 latency
>
> 256 (latent size: $32 \times 32$)
>
> | Method | Time | Params |
> | --- | --- | --- |
> | DiMSUM-L/2 | 2.20s | 460M |
> | DiT-L/2 | 3.80s | 458M |
>
> 512 (latent size: $64 \times 64$)
>
> | Method | Time | Params |
> | --- | --- | --- |
> | DiMSUM-L/2 | 2.86s | 461M |
> | DiT-L/2 | 4.78s | 459M |
>
> *Note: Previously, we measured our models's latency on two resolutions using the same device and GPU_ID (a single NVIDIA A100). However, for the result of our model upon 512x512, due to resource matter (occupied device), we used a different NVIDIA A100 (different device/GPU_ID), which could cause unfairness. We remesured using the same device/GPU_ID (same specs) for a fairer comparison.
>
> We appreciate your suggestion and have conducted additional benchmarks for 512x512 resolution images, following the same methodology used for 256x256 images in Table 2 of the rebuttal file. Our benchmarking process involves first warming up the model with 10 forward passes, then generating a single image (batch size = 1) 100 times, and finally calculating the average latency from these 100 runs.
>
> Note that the parameter change after changing image size is mainly due to the PatchEmbed layer of the architecture, which both models have.
>
> Q2. MEM and GFLOPs
>
> 256 (latent size: $32 \times 32$)
>
> | Method | MEM | GFlops |
> | --- | --- | --- |
> | DiMSUM-L/2 | 2.42G | 84.49 |
> | DiT-L/2 | 2.30G | 80.74 |
>
> 512 (latent size: $64 \times 64$)
>
> | Method | MEM | GFlops |
> | --- | --- | --- |
> | DiMSUM-L/2 | 2.46G | 337.48 |
> | DiT-L/2 | 2.34G | 361.14 |
>
> Given our time constraints, we focused our comparison on DiT-L/2, maintaining consistency with our latency comparison. The results, as shown in the table, reveal that DiMSUM-L/2's memory usage is slightly higher than its counterpart. This increase is expected, considering DiMSUM's slightly larger parameter count.
>
> Regarding GFlops, we acknowledge that for 256x256 images, DiMSUM produces about 4% more GFlops than DiT. However, an interesting trend emerges when we examine 512x512 images: DiMSUM's GFlops scaling is actually slower than DiT's. Consequently, at this higher resolution, DiT's GFlops exceed DiMSUM's by approximately 7%. This observation again aligns with the question 1 of reviewer **RLU7.**
>
> This observation aligns with the known quadratic complexity of transformers as sequence length increases. Our hybrid model mitigates this issue; the impact of attention blocks is reduced, while Mamba demonstrates its linear scaling complexity as the token count grows. This architectural choice allows DiMSUM to maintain efficiency at higher resolutions, offsetting the initial GFlops difference at 256 resolution.
>
> Q3-Q4.
>
> Thank you for asking the questions.
>
> In table 2, we use dopri5 to sample image from both models. We hypothesis that our architecture converges to less ODE-curvature solution compared to DiT architecture [1], [2]. This means that we could use less NFE to achieve the better image quality compared to DiT architecture. It is noted that when using dopri5, each sampling process could require different NFEs depending on the initial noise.
>
> Additionally, we acknowledge that our method requires more preprocessing before each MambaBlock, which may introduce latency to the inference speed. Furthermore, as shown in Table 2b, adding more Wavelet level processing costs only 0.01 GFLOPs. We also confirm that all preprocessing components are included in our measurements.
>
> [1] Pooladian, Aram-Alexandre, et al. "Multisample flow matching: Straightening flows with minibatch couplings." *arXiv preprint arXiv:2304.14772* (2023).
>
> [2] Lee, Sangyun, Beomsu Kim, and Jong Chul Ye. "Minimizing trajectory curvature of ode-based generative models." *International Conference on Machine Learning*. PMLR, 2023.
>
> Q5.
>
> To our knowledge, our method achieves state-of-the-art performance on several datasets, including CelebA and LSUN-Church. It's worth noting that while MDT and MaskDIT focused their benchmarks solely on ImageNet, our evaluation is more comprehensive, comparing against both transformer-based models and Mamba-based diffusion models such as DIFFUSSM and ZigMa.
>
> Regarding ImageNet, we acknowledge that our initial claim may cause confusions. We will revise our statement to more accurately reflect our findings: "Our method demonstrates superior results compared to DiT and DIFFUSSM, achieving faster training convergence and delivering high-quality outputs."
>
> It's important to highlight that both MaskDiT and MDT report their results using significantly larger models (-XL/2, approximately 675M parameters), while our results are based on a smaller model (-L/2, 460M parameters). Despite this size difference, our model achieves competitive performance. In fact, DiMSUM-L/2-G attains a slightly better FID score (2.11) than MaskDiT-G (2.28).
>
> We acknowledge your concerns as they are critical for the paper’s clarity and improvement. We will revise them in our manuscript.

---

> > ### Comment · Reviewer_C5pc · 2024-08-13
> >
> > 1. Could authors confirm that the proposed method is faster mainly because of using less NFE in sampling? And will it be slower if both methods are used under the same NFE?
> >
> > 2. How many NFEs are used concretely for the tables just newly provided for each method (DiT and DiMSUM)?
> >
> > This is to ensure the understanding that the advantage of speed comes from the Mamba thing or just using fewer NFEs. Reading the whole paper, I just had in my mind that the model with the Mamba hybrid Transformer is faster than the pure Transformer, but it is not if they just use the same NFEs. And of course, their proposed method satisfied the need for more Memory and more parameters used.

---

> > > ### Author Response · Authors · 2024-08-13
> > >
> > > Thanks for your quick response! We provides our answers to your questions as follows:
> > >
> > > Q1.
> > >
> > > Observing the Memory and GFLOPS table above, it's true to claim DiMSUM-L/2 shows slower inference speed compared to its counterpart for 256x256 images using the same NFE (due to bigger GFLOPS), however, there are two crucial points to consider:
> > >
> > > 1. Scaling Efficiency:
> > > When we increase the image size to 512x512, as evident from the Memory and GFLOPS table, our model actually requires fewer GFLOPS at this higher resolution, thanks to its slower latency scaling which we mentioned above. Consequently, for 512x512 images and larger, DiMSUM-L/2 would outperform its counterpart in speed given the same NFE.
> > > 2. Adaptive Sampling Efficiency:
> > > We employ the dopri5 ODE adaptive solver for sampling from both models. This solver dynamically adjusts the NFE based on the initial noise and diffusion model characteristics, using the minimum NFE necessary to achieve optimal image quality. Notably, DiMSUM requires fewer NFE to meet the dopri5 stopping condition while still achieving a significantly better FID score than DiT. **We hypothesize that our proposed hybrid architecture converges to a better solution with less curvature, enabling high-fidelity image production with fewer NFEs.**
> > >
> > > With these two points, we emphasize upon the importance of our proposed architecture, rather than just the benefits given by the flow matching framework.
> > >
> > > Q2. About the exact number of NFE:
> > > Since dopri5 requires different NFEs depending on initial noise, we sample 100 random noises and generate images for both models using dopri5 to measure the NFE. The average NFEs for DIMSUM and DIT at resolution 256 are 61 and 143. For resolution 512, the average NFEs for DIMSUM and DIT are 82 and 138.

---

> > > > ### Author Response · Authors · 2024-08-14
> > > > **Thank you.**
> > > >
> > > > We sincerely thank you for your thorough review and your review is insightful and helpful to our paper. We hope our answer could resolve your concern. We wish you a good Neurips.

---

> ### Author Response · Authors · 2024-08-13
> **Clarification of DiT term in the rebuttal**
>
> This section was included to the global response above, however, we include it here for easy reference.
>
> We'd like to clarify that when we refer to DiT-L/2 in the rebuttal, we specifically mean the DiT-L/2 model trained using the flow matching framework like SiT. This choice ensures a fair comparison of inference speed.
>
> This comparison is more equitable because our DiMSUM model also employs the flow matching framework. Crucially, the sampling methods between traditional diffusion methods and flow matching methods differ significantly. By comparing DiMSUM to the flow matching-trained version of DiT (SiT), we ensure that both models use the same sampling approach during inference.
>
> By aligning sampling methods, we can accurately evaluate inference speed differences, ensuring that any variations are due to architectural choices rather than differences in sampling or training processes.
>
> We also sorry for the typo in table 2 of the rebuttal PDF, the parameters of DiMSUM-L/2 should be 460M, not 480M.

---

### Official Review · Reviewer_nhSz · 2024-06-30

**Soundness:** 4
**Presentation:** 4
**Contribution:** 4
**Rating:** 8
**Confidence:** 5

**Summary:**

This paper introduces a novel architecture for diffusion models that leverages spatial and frequency information to emphasize local features in image generation. It integrates wavelet transformation into the state-space networks, such as Mamba, to enhance the awareness of local structures in visual inputs. The outputs are fused with original Mamba outputs using a cross-attention fusion layer to optimize order awareness, crucial for image generation quality. A globally-shared transformer is added to boost Mamba's performance. The method achieves state-of-the-art results on standard benchmarks, with faster training convergence and high-quality outputs.

**Strengths:**

- Overall, I appreciate this paper. It selects the most promising technical stacks—combining flow matching and Mamba—to address generative tasks.
- The paper is well-organized, featuring valuable methodology design and experimental ablations.
- It's interesting to see that the community has started considering state exploration in Mamba, as indicated in Line245. The initialization of the state is critical. We need to examine the code to understand how to implement it efficiently. See https://github.com/state-spaces/mamba/issues/101 for more details.

**Weaknesses:**

- In Line 106, I'm unsure why sigma brings extra computation when the scan path is larger, given that the computation is always fixed.
- In Line105, it's important to note that flow matching has been utilized in various domains to capture the reader's interest. E.g., boosting diffusion[1], depth estimation[2], motiom[3], even text generation[4].
- In Line 176, I'm curious as to why only B, C, and delta are time-dependent. Why isn't A?
- In Line183, One paper is missed. [5]
- In Fig 3a, what does the notation on ZigMa mean?
- Regarding line 261, what's the rationale behind sharing those parameters of transformers? Are there any previous works indicating its effectiveness?
- In Table 1, it would be better to display training steps. Researchers of generative models tend to compare using training steps rather than epochs.
- In Line342, for sweep4, considering four forwards in a single-forward usually leads to a worse training speed (iter/s) and increased memory usage. It would be interesting to compare these two aspects with other methods.
- In Fig5, is there any reference to the jpeg-8 scan path? How does it defined?

Suggestions:

- The previously mentioned figure or table should be placed first.

[1]. Boosting Latent Diffusion with Flow Matching

[2]. DepthFM: Fast Monocular Depth Estimation with Flow Matching

[3]. Motion Flow Matching for Human Motion Synthesis and Editing

[4]. Flow Matching for Conditional Text Generation in a Few Sampling Steps

[5], Latent Space Editing in Transformer-based Flow Matching

Overall, I think the exploration of using Mamba and flow matching is the right way to go, while I have some technical concerns about it. I am eager to hear the authors’ feedback.

**Questions:**

as above

**Limitations:**

as above

---

> ### Author Rebuttal · Authors · 2024-08-07
>
> 1. In Line 106, I'm unsure why sigma brings extra computation when the scan path is larger, given that the computation is always fixed.
>
> 2. In Line105, it's important to note that flow matching has been utilized in various domains to capture the reader's interest. E.g., boosting diffusion[1], depth estimation[2], motiom[3], even text generation[4].
>
> 3. In Line183, One paper is missed.
>
> For Q1,2,3, we admit that our wording is incorrect in this context and revise the sentence as "In this paper, we show that too many scanning orders, e.g., sweep-8 and zigzag-8, may introduce excessive information and lead to worse performance compared to sweep-4". We will cite the suggested papers.
>
> 4. In Line 176, I'm curious as to why only B, C, and delta are time-dependent. Why isn't A?
>
> We believe that the authors of Mamba would have a much better answer here. From our point of view, this implementation is made purely to prefer simplicity without sacrificing much performance. For more details, reviewers can check the interpretation in 3.5.2 of the Mamba paper. Simply put, time-dependent delta $\Delta_t$ is sufficient to ensure the selection mechanism for $\bar{A}$ via $\exp(A * \Delta_t)$.
>
> 5. In Fig 3a, what does the notation on ZigMa mean?
>
> We are sorry for the missing information; we will update the caption in the revised version. In this case, $\textdagger$ is our reproduced result based on Zigma's official code, and $\textdaggerdbl$ is an adopted result from LFM paper.
>
> 6. Regarding line 261, what's the rationale behind sharing those parameters of transformers? Are there any previous works indicating its effectiveness?
>
> As mentioned in L260, we drew inspiration from Zamba for the globally shared attention block. The primary purpose of this component is to reduce the model's parameter count while maintaining the competitive performance of the model by capturing the global dependencies between input tokens. Besides, Zigma also adopts a hybrid mamba-transformer architecture in the design to integrate the text condition for text-to-image generation effectively.
>
> 7. In Table 1, it would be better to display training steps. Researchers of generative models tend to compare using training steps rather than epochs.
>
> We acknowledge your comment and update Table 1 in the attached file.
>
> 8. In Line342, for sweep4, considering four forwards in a single-forward usually leads to a worse training speed (iter/s) and increased memory usage. It would be interesting to compare these two aspects with other methods.
>
> Actually, in our implementation, we don't run all scans in one forward (such as in VMamba), but we run each order alternatively following Mamba-ND and Zigma paper. Specifically, each Mamba block handles only one direction, and the four scanning orders in sweep4 are evenly distributed throughout the network's depth (number of layers). Sweep4, for example, has four scanning orders: (1) left-to-right, (2) right-to-left, (3) top-to-bottom, and (4) bottom-to-top. Assuming the network has only four layers, layer 1 handles (1), layer 2 handles (2), and so on.
>
> 9. In Fig5, is there any reference to the jpeg-8 scan path? How does it defined?
>
> As mentioned in L330, we derived the JPEG-8 scanning order from the scanning order of the JPEG compression algorithm, as described in "A fast JPEG image compression algorithm based on DCT". This also was significantly influenced by the Zigma paper, a pioneering work that put effort into a more sophisticated scanning order and achieved competitive results. Building upon their insights, we extended the concept to the original JPEG scanning orders. We acknowledge the valuable contribution of the Zigma paper and will provide a more detailed explanation of our scanning order development in the camera-ready version for improved clarity.
>
> 10. The previously mentioned figure or table should be placed first.
>
> As mentioned in the global response, we thank the reviewer for your attention to detail, which is extremely vital to enhance the clarity and completeness of the paper. We will try our best to reorder the figures and tables so that it would appear before being mentionned.

---

> > ### Comment · Reviewer_nhSz · 2024-08-12
> > **reply**
> >
> > Thanks for the author's reply. I have also read the other reviewer's opinion.
> >
> > My concerns are fully resolved, so I will increase my score to 8 to reflect this.
> >
> > I encourage the authors to consolidate the results on ImageNet to compare with related baselines with Zigma.

---

> > > ### Author Response · Authors · 2024-08-12
> > >
> > > Thank you for raising your concerns as well as for the detailed review of our paper's presentation. Addressing those definitely improves our paper generally. We will further follow your valuable recommendation!

---

> > > > ### Author Response · Authors · 2024-08-14
> > > > **Thank you.**
> > > >
> > > > We sincerely thank you for your thorough review and your review is insightful and helpful to our paper. We hope our answer could resolve your concern. We wish you a good Neurips.

---

### Official Review · Reviewer_eitf · 2024-07-08

**Soundness:** 3
**Presentation:** 3
**Contribution:** 3
**Rating:** 5
**Confidence:** 4

**Summary:**

This paper proposes a novel Mamba-based diffusion model, DiMSUM, which leverages both spatial and frequency information to enhance visual generation capabilities. Specifically, DiMSUM applies wavelet transformation to the input signals, decomposing them into wavelet subbands. By employing a query-swapped cross-attention technique, DiMSUM effectively integrates spatial and frequency information. Furthermore, DiMSUM incorporates several transformer blocks into the Mamba model, enriching the global context features.

**Strengths:**

1.	The paper is well-motivated. It observes the difficulties caused by the manually-defined scanning orders in tradition Mamba-based diffusion models, and uses transformation in the frequency domain to mitigate this problem.
2.	The paper is well-written and easy to follow.

**Weaknesses:**

1.	Since the authors argue the incorporation of frequency information can mitigate the problem caused by the scanning order, I urge the authors to conduct an experiment to verify whether the generation performance of DiMSUM model is sensitive to the scanning order.
2.	Stronger baselines (such as [1]) and the experiments on more datasets (such as conditional generation on MSCOCO) and resolutions are required.
3.	From Table 2(a), it seems that the Wavalet transformations would harm the performance (i.e., FID and recall), which cannot demonstrate the effectiveness of the proposed method.


[1] Chen, Junsong, et al. "Pixart-$\alpha $: Fast training of diffusion transformer for photorealistic text-to-image synthesis." arXiv preprint arXiv:2310.00426 (2023).

**Questions:**

See the weaknesses part.

Meanwhile, I recommend the authors to provide sufficient experiments to support your method. And I would consider increasing my score if the above concerns can be addressed.

**Limitations:**

See the weaknesses part.

All of my concerns have been addressed. I will increase my score.

---

> ### Author Rebuttal · Authors · 2024-08-07
>
> 1. Since the authors argue the incorporation of frequency information can mitigate the problem caused by the scanning order, I urge the authors to conduct an experiment to verify whether the generation performance of DiMSUM model is sensitive to the scanning order.
>
> To substantiate our claims regarding the efficiency of frequency information, we conducted a comprehensive ablation study. This study utilized four scanning orders: (1) bidirection, (2) jpeg-8, (3) sweep-8, and (4) zigzag-8. We trained models on CelebA-HQ at 256x256 resolution for 250 epochs, comparing performance when applying these scanning strategies to: a) spatial domain only or b) both spatial and frequency domains
>
> Table 3 of rebuttal file presents the results of this ablation study. Notably, integrating frequency domain information across all four scanning strategies led to significant performance improvements. This consistent enhancement across various scanning methods provides strong evidence for the effectiveness of our approach in leveraging frequency information.
>
> 2. Stronger baselines (such as Pixart-alpha) and the experiments on more datasets (such as conditional generation on MSCOCO) and resolutions are required.
>
> We appreciate the suggestion to explore text-to-image (T2I) generation. However, training competitive text-to-image models presents significant challenges:
>
> - Computational constraints: As a small lab, we lack the extensive resources required for training large-scale T2I models.
> - Dataset availability: Recent closure of datasets like LAION-5B limits access to suitable training data for T2I tasks and also would be hard to compare fairly with previous works using those datasets.
>
> It's worth noting that focusing on label-to-image tasks, as we have done, is common in diffusion model research like DiT, EDM, and Consistency Models. Many influential papers in this field have established their contributions through experiments on label-to-image before expanding to text-to-image.
>
> 3. From Table 2(a), it seems that the Wavalet transformations would harm the performance (i.e., FID and recall), which cannot demonstrate the effectiveness of the proposed method.
>
> We hypothesize that spatial and frequency signals are not aligned and require careful integration to leverage their information. Naively fusing these domains (e.g., by concatenation) can damage performance due to conflicting or misaligned information.
>
> To address this challenge, we proposed a more sophisticated fusion method using Cross-Attention layers between these spaces. This approach enables the model to effectively combine information from both domains, leveraging their strengths while mitigating potential conflicts. Hence, this fusion technique can enhance the FID from 5.87 to 4.92 in Table 2c submission, contributing to the SoTA result of our method.

---

> > ### Comment · Reviewer_eitf · 2024-08-10
> >
> > Thanks for the feedback. All of my concerns have been addressed. I will increase my score.

---

> > > ### Author Response · Authors · 2024-08-10
> > >
> > > We sincerely thank you for the insightful review and supportive feedback to make the paper more complete. Your support is highly appreciated, and we're glad that our responses have addressed your concerns.

---

> > > > ### Author Response · Authors · 2024-08-14
> > > > **Thank you.**
> > > >
> > > > We sincerely thank you for your thorough review and your review is insightful and helpful to our paper. We hope our answer could resolve your concern. We wish you a good Neurips.

---

### Official Review · Reviewer_RLU7 · 2024-07-09

**Soundness:** 2
**Presentation:** 3
**Contribution:** 2
**Rating:** 4
**Confidence:** 4

**Summary:**

This paper proposes to employ a Mamba-transformer hybrid framework with wavelet-spatial scanning for image generation. The authors claim that the scanning in the frequency domain can model the long-range relations of tokens in 2D spatial space. The experiments and ablation studies have shown the effectiveness of the proposed method.

**Strengths:**

- This paper is easy to follow.

- The implementation details and the codes are included in this submission.

- The performance on multiple datasets and sufficient ablation studies have verified the effectiveness of the proposed method.

**Weaknesses:**

- The proposed method is too complex. Scanning the image and frequency space makes the framework complicated. The authors further introduced transformer blocks in the mamba framework, which added more complexity to the entire method.

- Why do the authors mention "manually-defined scanning orders" in their abstract? This issue is not discussed in the subsequent contents of the paper. Moreover, it cannot be addressed by the scanning in the frequency domain. The representation of an image in the frequency domain also has two dimensions, so the challenges for the scanning on the frequency domain should be the same as the scanning on the image space. Moreover, the scanning directions of the proposed method are also manually defined, even a window scanning is involved.

- The comparison of the real inference speed between the proposed framework and transformers is not included in this paper.

- Although the authors claim that they propose a Mamba-based diffusion model. However, this model still includes many transformer blocks, so this is a hybrid model rather than a Mamba model. Some claims in this paper like ''mamba-based'' or ''Mamba models'' should be revised. Moreover, is the spatial-frequency scan also powerful on the pure transformer model? Does the Mamba-transformer beat the pure transformer model given the same input setting?

- The title of this paper claims that the proposed method is scalable. However, the parameter counts of the proposed model is smaller than 1.0B, so this title may not be proper.

**Questions:**

- In Figure 3(d), why does DiT perform so poorly on CelebA-HQ? Are there any bugs? DiT works very well on many large-scale datasets.

**Limitations:**

Although this paper uses public datasets, at least a section for social impact should be included in this paper. Moreover, this papers does not discuss the limitations of the proposed method.

---

> ### Author Rebuttal · Authors · 2024-08-07
>
> 1. Our architecture might comprise many components such as Mamba, Transformer, and frequency processing. However, each proposed component is well-motivated and vital to the overall framework, as shown in Table 2a (in submission). The simple Mamba network, without frequency processing and transformer blocks, cannot surpass the performance of the transformer. As shown in section 2.1 submission, the scanning order of the Mamba block has a significant effect. To mitigate its impact, we integrate wavelet frequency with window scanning into architecture. As explained in the global response, wavelet scanning, along with window scanning, can better capture the global information of each wavelet subband, enhancing the overall global information across the whole image. In addition, our model could benefit from learning both high and low frequency [1]. With wavelet frequency and window scanning, our framework reduces FID from 5.27 to 4.92 for CelebA-HQ (Table 2e in submission). Furthermore, recently, hybrid architectures of Transformer and Mamba [2, 3] have demonstrated SoTA performance in NLP tasks. Motivated by these works, we measure the performances of hybrid Transformer-Mamba in vision generative. As shown in Table 2f submission, both independent and shared transformer improves performance, indicating hybrid model's effectiveness.
>
> [1]: Hao Phung, Quan Dao, and Anh Tran. "Wavelet diffusion models are fast and scalable image generators." In CVPR 2023.
>
> [2]: Zyphra unveils zamba: A compact 7b ssm hybrid model, 2024. https://www.zyphra.com/zamba
>
> [3]: O. Lieber et al. Jamba: A hybrid transformer-mamba language model. arXiv preprint arXiv:2403.19887, 2024
>
> 2. We acknowledge that the presentation of our paper should be clearer and appreciate your feedback. To clarify, the term "manually-defined scanning orders" refers to the heuristic scanning order in the Mamba blocks. In Section 2.1, from lines 94 to 107 (in submission), the scanning order of Mamba is defined as hand-crafted in previous works. Throughout the paper, we mentioned some of these hand-crafted orders, including "bi", "window", "sweep-4/8", and "zigzag-8". These scanning orders are illustrated in Figs 2, 5, and 6 in submission.
> Although our approach does not completely address the impact of scanning order, our window scanning strategy in wavelet space could efficiently capture the global information within each frequency sub-bands which improves global content of generated image, as explained in global response. Please refer to table 3 (in rebuttal file), our wavelet and window scan component successfully reduce FID score compared to spatial scanning only. Therefore, we consider our frequency scanning method to be synergistic with image space scanning, as it effectively handles the global information of generated image. Besides, completely resolving the limitations of manually-defined scanning is an interesting topic for future research.
>
> 3. DIMSUM-L/2 achieves 2.2 seconds latency compared to DIT-L/2 with 3.8 seconds (see table 2 in rebuttal file).
>
> 4. In terms of model's type, we would like to revise the terms "Mamba-based" and "Mamba model" to "a hybrid Mamba-Transformer network" for better comprehension. In terms of spatial-frequency scan for transformer, we modify DiT architecture to further incorporate wavelet frequency information similar to DIMSUM. After training spatial-frequency transformer on CelebA-HQ for 1000 epoch, we found that the spatial-frequency transformer failed to learn the human face distribution. The output model only generates noise-pattern images. More rigorous analysis of model is beyond the scope of paper and we would like to leave it for future investigation.
>
> 5. We acknowledge that the use of term 'scalable' may have caused some confusion. In the broader context, deep learning’s scalability may refer to capacity to handle various datasets and computational resources while producing correct results in an acceptable period of time [1].
>
> Our model aligns well with this definition of scalability. Our experiments demonstrated that:
> - The model adapts to multiple datasets with minimal hyperparameter tuning (similar to DiT paper).
> - It achieves competitive performance metrics compared to other methods.
> - Inference speed is also faster, as shown in table 2 (in rebuttal file).
>
> Our SoTA results are achieved with a parameter count comparable (or even smaller) to existing models. Refer to Figure 3a and 4a (in submission) and Table 1 (in rebuttal file). This suggests substantial room for further enlargement of our model's parameters, which we anticipate will yield even better improvements across various and bigger datasets (see table 4 in rebuttal file).
>
> 6. Our plot intentionally stops at epoch 300, demonstrating the model's capability to converge faster than other methods. DiT does perform well on CelebA-HQ but takes more than 500. Figure 4 (in rebuttal file) illustrates the convergence of DiT at the later epoch. It's noted that DiT and LFM (in figure 3d submission) use the same DiT-L/2 architecture. While DiT uses diffusion loss, LFM uses flow matching. LFM converge faster than DiT. However, our model with flow matching loss, demonstrates even faster convergence than LFM, indicating our architecture enhances convergence rate.
>
> 7. Social Impact section will be added in revised paper (see global response).

---

> > ### Comment · Reviewer_RLU7 · 2024-08-13
> >
> > Thank you for your reply. Your rebuttal has solved most of my concerns, but some are still not addressed.
> > The remaining concerns almost overlap with Reviewer C5pc's new concerns.
> >
> > For example,
> >
> > (1) why your mamba-transformer is faster than DiT? The rebuttal shows that DIMSUM is faster than DiT on ImageNet 256 (256 tokens considering the downsampler of the pre-trained autoencoder), however, a single Mamba module is faster than a transformer module only when the number of tokens is greater than 1000 (please refer to Fig. 8 in [1]);
> >
> > (2) the performance of DiMSUM after longer training on ImageNet 256 is 2.11 which is worse than SiT (FiD=2.06) [2], though both of these methods use flow matching. Thus, the performance gain of Mamba remains ambiguous (at least marginal).
> >
> > [1] Gu A, Dao T. Mamba: Linear-time sequence modeling with selective state spaces[J]. arXiv preprint arXiv:2312.00752, 2023.
> >
> > [2] Ma N, Goldstein M, Albergo M S, et al. Sit: Exploring flow and diffusion-based generative models with scalable interpolant transformers[J]. arXiv preprint arXiv:2401.08740, 2024.

---

> ### Author Response · Authors · 2024-08-13
>
> Thank you for your response! We provide our answers below:
>
> (1) Why faster?
>
> We agree with the statement “a single Mamba module is faster than a transformer module only when the number of tokens is greater than 1000”.  In the provided table below, we can see that Gflops of DIMSUM with 256 token is higher than Gflops of DiT. The reason why our model take less time to generate image is that we use dopri5 to produce image at inference time. We conjecture that DIMSUM could converges to less ODE-curvature solution compared to DiT architecture [1], [2]. Therefore, our architecture use less NFE to achieve the better image quality compared to DiT architecture (this is also indicated by our FID result on several benchmarks like CelebA-HQ and Church). Moreover, with resolution 512 (1024 tokens), we can see that our architecture achieves lower Gflops and inference time. This demonstrate the potential use of DIMSUM in larger benchmark like text-to-image which has larger resolution.
>
> [1] Pooladian, Aram-Alexandre, et al. "Multisample flow matching: Straightening flows with minibatch couplings." *arXiv preprint arXiv:2304.14772* (2023).
>
> [2] Lee, Sangyun, Beomsu Kim, and Jong Chul Ye. "Minimizing trajectory curvature of ode-based generative models." *International Conference on Machine Learning*. PMLR, 2023.
>
> 256 (latent size: $32 \times 32$)
>
> | Method | Time | Params | GFlops |
> | --- | --- | --- | --- |
> | Ours-L/2 | 2.20s | 460M | 84.49 |
> | DiT-L/2 | 3.80s | 458M | 80.74 |
>
> 512 (latent size: $64 \times 64$)
>
> | Method | Time | Params | GFlops |
> | --- | --- | --- | --- |
> | Ours-L/2 | 2.86s | 461M | 337.48 |
> | DiT-L/2 | 4.78s | 459M | 361.14 |
>
> *Note: In the previous edition, we measured the time latency of all models on two resolutions using the same device and GPU_ID (a single NVIDIA A100). However, for the result of our model upon 512x512, due to resource matter (occupied device), we used a different NVIDIA A100 (different device/GPU_ID). We acknowledged that changing devices could cause unfairness. We fixed it by remeasuring using the same device/GPU_ID (same specs) for a fairer comparison. This issue doesn't affect GFLOPS since we did this separately and already used the same device/GPU_ID. Sorry for the inconvenience.
>
> (2) Compare with SiT
>
> Thank you for noting the SiT model comparisons. To clarify, the SiT model in Table 1 of our paper (FiD=2.15) and the SiT model with FiD=2.06 are the same but with different samplers. We reported the FiD=2.15 result for a fairer NFE comparison, which we'll explain further below.
>
> The SiT authors used two sampling methods for their model. Table 9 of the SiT paper shows results for SiT-XL (cfg=1.5, ODE) with FiD=2.15 and SiT-XL (cfg=1.5, SDE:σ_t) with FiD=2.06. The latter, achieved with a 250 NFE Heun (SDE) solver, is computationally equivalent to a 500 NFE Euler solver, as the Heun solver requires an additional model forward pass per step.
>
> Given the substantial computational requirements and our limited remaining time, evaluating ImageNet using this method is not feasible during the rebuttal phase. However, our comparison using SiT-XL (cfg=1.5, ODE) remains valid. Insights from SiT’s section 3.5 and Table 9 suggest that DiMSUM, using the same flow matching and training framework, would likely see a similar performance boost with the SDE sampler. We will conduct this evaluation and include the results in the revised version, with clear specification of the sampling method for full transparency.
>
> Lastly, we'd like to highlight that our model is considerably smaller than the SiT model (460M vs 675M parameters). This size difference indicates the model scalability for improvement in our approach, as shown by the parameter scaling ablation in Table 4 of our rebuttal. This study reveals that the upscaled DiMSUM-XL/2 slightly outperforms the L/2 version in FID (3.76 vs 3.45).
>
> We acknowledge your concerns as they are critical for the paper’s clarity and improvement. We will revise them in our manuscript.

---

> ### Author Response · Authors · 2024-08-13
> **Clarification of DiT term in the rebuttal**
>
> This section was included to the global response above, however, we include it here for easy reference.
>
> We'd like to clarify that when we refer to DiT-L/2 in the rebuttal, we specifically mean the DiT-L/2 model trained using the flow matching framework like SiT. This choice ensures a fair comparison of inference speed.
>
> This comparison is more equitable because our DiMSUM model also employs the flow matching framework. Crucially, the sampling methods between traditional diffusion methods and flow matching methods differ significantly. By comparing DiMSUM to the flow matching-trained version of DiT (SiT), we ensure that both models use the same sampling approach during inference.
>
> By aligning sampling methods, we can accurately evaluate inference speed differences, ensuring that any variations are due to architectural choices rather than differences in sampling or training processes.
>
> We also sorry for the typo in table 2 of the rebuttal PDF, the parameters of DiMSUM-L/2 should be 460M, not 480M.

---

> > ### Author Response · Authors · 2024-08-14
> > **Thank you.**
> >
> > We sincerely thank you for your thorough review and your review is insightful and helpful to our paper. We hope our answer could resolve your concern. We wish you a good Neurips.

---

### Author Rebuttal · Authors · 2024-08-07

We address the reviewers' comments below, referring to them as: R1(RLU7), R2(eitf), R3(nhSz), R4(C5pc). We sincerely thank all reviewers for their valuable feedback. We appreciate the positive comments on clear writing and good flow (R1, R2, R3), well-defined motivation to investigate and integrate Mamba with Transformer (R2, R3, R4), our novel approach of integrating frequency space into Mamba (R4), and thorough and extensive experiments (R1, R4).

**Scanning in Frequency Space**:

We would like to further elaborate on the motivation and effectiveness of our frequency scanning method. Mamba-based approaches in diffusion models often struggle with efficiently scanning patches while maintaining local and global 2D-spatial information. Although several works have proposed sophisticated scanning methods to address this issue, these approaches are complex and computationally expensive. Conversely, some methods have adopted local scanning (i.e., windowing) strategies [3] to improve model latency and throughput. Still, these often underperform compared to previously mentioned scanning methods as it is limited to the dependencies of nearby-pixels within window.

DiMSUM addresses these challenges by breaking the original image into frequency wavelet subbands (each subband is half of resolution of image). This approach is efficient to capture long-range frequency while preserving information across different subbands. We redesigned the window scanning, where each window corresponds to a subband of the frequency space (Fig 1 rebuttal PDF). Consequently, each window captures low/high frequency signal of original image. Since each frequency sequence is 1/4 of image sequence, negative impact of scanning is reduced. As the model progresses through different subbands, it incorporates spatial information represented at various low-to-high frequencies, adding valuable context to the process. This key difference distinguishes our method from traditional window scanning in image space.

We conducted two key ablation studies to validate this:

1. Table 3 (rebuttal PDF) shows models using both frequency and spatial domains consistently outperform spatial-only models across various scanning orders, demonstrating the robustness and effectiveness of frequency domain intergration.
2. Table 2e (submission) confirms our window-scanning strategy's efficacy in the frequency domain, achieving the best FID of $4.92$ on CelebA-HQ at $256 \times 256$ resolution.

These studies substantiate the effectiveness of our frequency domain integration and scanning strategy.

**Scalable term**

 We acknowledge that the use of the term 'scalable' may have caused some confusion, given its current association with model parameter scaling in the LLM-dominated landscape. However, in the broader context, machine learning/ deep learning algorithms’ scalability may refer to their capacity to handle bigger datasets and computational resources while producing correct results in an acceptable period of time.

Our model aligns well with this definition of scalability. Our experiments demonstrated that:
 - The model adapts to multiple datasets with minimal hyperparameter tuning (similar to DiT paper).
-  It achieves competitive performance metrics compared to other methods.
- Inference speed is also faster, as shown in table 2 (in rebuttal file).

Our SoTA results are achieved with a parameter count comparable (or even smaller) to existing models. Refer to Figure 3a and 4a (in submission) and Table 1 (in rebuttal file). This suggests substantial room for further enlargement of our model's parameters, which we anticipate will yield even greater improvements across various and bigger datasets (see table 4 in rebuttal file).

**Presentation**:

We appreciate the reviewers' suggestions for improving our presentation. We will address the following in our revised version:
1. Inconsistent/Misunderstanding terms (e.g., "manually-defined scanning orders," "Mamba-based" (R1)
2. Addition of social impacts and limitations sections (R1).
3. Claim about the computation of larger scan path (R3).
4. Missing citations and annotation definitions in captions (R3).
5. Figures and tables orderings (R3).
6. Updated ImageNet table with training iterations instead of epochs as suggested by (R3) and the new results of our method when trained for the same number of iterations as DIFFUSSM. Additionally, we apologize for the typo regarding the training epochs for DIFFUSSM. It should be corrected from 1.4K to 515.

**Social impacts and limitations**:

We believe that our proposed network advances the architectural design of state-space models for image generation. This model can be extended to various tasks, such as large-scale text-to-image generation and multimodal diffusion. While there is a risk that our architecture could be misused for malicious purposes, posing a social security challenge, we are confident that this risk can be mitigated with the recent development of security-related research. Hence, the positives can outweigh the negative ones, rendering the concern minor.

While our method outperforms other diffusion baselines in generation quality and training convergence, we acknowledge areas for improvement. These include enhancing multiscale feature paradigms and multi-scale diffusion loss robustness [1, 2], and addressing manually-defined scanning orders. We're also intrigued by recent masked-training diffusion methods (e.g., MaskDiT, MDT), which reduce computational requirements and potentially improve global context learning. Integrating these orthogonal approaches with our work could yield further improvements.

[1] Crowson, et al. "Scalable high-resolution pixel-space image synthesis with hourglass diffusion transformers." In ICML, 2024.

[2] Hoogeboom, et al. "simple diffusion: End-to-end diffusion for high resolution images." In ICML, 2023.

[3] Huang, et al. "Localmamba: Visual state space model with windowed selective scan." In Arxiv, 2024.

---

> ### Author Response · Authors · 2024-08-13
>
> We'd like to clarify that when we refer to DiT-L/2 in the rebuttal, we specifically mean the DiT-L/2 model trained using the flow matching framework like SiT. This choice ensures a fair comparison of inference speed.
>
> This comparison is more equitable because our DiMSUM model also employs the flow matching framework. Crucially, the sampling methods between traditional diffusion methods and flow matching methods differ significantly. By comparing DiMSUM to the flow matching-trained version of DiT (SiT), we ensure that both models use the same sampling approach during inference.
>
> By aligning sampling methods, we can accurately evaluate inference speed differences, ensuring that any variations are due to architectural choices rather than differences in sampling or training processes.
>
> We also sorry for the typo in table 2 of the rebuttal PDF, the parameters of DiMSUM-L/2 should be 460M, not 480M.

---

### Comment · Area_Chair_27KU · 2024-08-09
**Read the rebuttal and discuss with the authors**

Hi Reviewers,

The authors have submitted their rebuttal responses addressing the concerns and feedback you provided. We kindly request that you review their responses and assess whether the issues you raised have been satisfactorily addressed.

If there are any areas where you believe additional clarification is needed, please do not hesitate to engage in further discussion with the authors. Your insights are invaluable to ensuring a thorough and constructive review process.

Best
AC

---

### Decision · Program_Chairs · 2024-09-25

**Decision:**

Accept (poster)

**Comment:**

The paper received mixed reviews from four reviewers: one strong accept, one borderline accept, and two borderline rejects. All reviewers appreciated the novelty and effectiveness of the proposed state-space architecture for diffusion models. The authors effectively addressed most of the reviewers' concerns during the rebuttal. However, the two reviewers who gave borderline rejects still have some concerns regarding inference speed and performance with longer training epochs. The authors provided further explanation, but these two reviewers did not respond to the authors' additional explanations.

After reviewing the paper and the rebuttal, AC finds the authors' explanations satisfactory. Given the overall contribution and the positive aspects highlighted, AC believes the paper is worthy of acceptance and should be shared with the research community. This decision has been discussed with SAC. At the same time, AC encourages the authors to add some important rebuttal content into the final camera-ready version and polish their paper by following reviewers' suggestions.